# Development and Validation of a New In-Situ Technique to Measure Total Gaseous Chlorine in Air

Teles C. Furlani[1], RenXi Ye[1], Jordan Stewart[1,2], Leigh R. Crilley[1], Peter M. Edwards[2], Tara F. Kahan[3], Cora J. Young[1*]

[1] Department of Chemistry, York University, Toronto, Canada

[2] Department of Chemistry, University of York, York, UK

[3] Department of Chemistry, University of Saskatchewan, Saskatoon, Canada

*Correspondence to: youngcj@yorku.ca

**Abstract**

Total gaseous chlorine ($TCl_g$) measurements can improve our understanding of unknown sources of Cl to the atmosphere. Existing techniques for measuring $TCl_g$ have been limited to offline analysis of extracted filters and do not provide suitable temporal information on fast atmospheric process. We describe high time-resolution in-situ measurements of $TCl_g$ by thermolyzing air over a heated platinum (Pt) substrate coupled to a cavity ring-down spectrometer (CRDS). The method relies on the complete decomposition of $TCl_g$ to release Cl atoms that react to form HCl, for which detection by CRDS has previously been shown to be fast and reliable. The method was validated using custom organochlorine permeation devices (PDs) that generated gas-phase dichloromethane (DCM), 1-chlorobutane (CB), and 1,3-dichloropropene (DCP). The optimal conversion temperature and residence time through the high-temperature furnace was 825 °C and 1.5 seconds, respectively. Complete conversion was observed for six organochlorine compounds, including alkyl, allyl, and aryl C-Cl bonds, which are amongst the strongest Cl-containing bonds. The quantitative conversion of these strong C-Cl bonds suggests complete conversion of similar or

weaker bonds that characterize all other $TCl_g$. We applied this technique to both outdoor and indoor
environments and found reasonable agreements in ambient background mixing ratios with the sum
of expected HCl from known long-lived Cl species. We measured the converted $TCl_g$ in an indoor
environment during cleaning activities and observed varying levels of $TCl_g$ comparable to previous
studies. The method validated here is capable of measuring in-situ $TCl_g$ and has a broad range of
potential applications.
**1. Introduction**

Chlorine (Cl) containing compounds in the atmosphere can impact air quality, climate, and

health (Saiz-Lopez and Von Glasow, 2012; Simpson et al., 2015; Massin et al., 1998; White and
Martin, 2010). Gaseous chlorinated compounds are either organic (e.g., dichloromethane,
chloroform, and carbon tetrachloride) or inorganic (e.g., $Cl_2$, HCl, and $ClNO_2$), with inorganic Cl
being more reactive under most atmospheric conditions. In this work, total gaseous Cl ($TCl_g$) refers
to all gas-phase Cl-containing species weighted to their Cl content, including both inorganic and
organic species. While groups of chlorinated species are often considered based on reactivity
considerations (e.g., reactive chlorine, $Cl_y$), $TCl_g$ includes all molecules that contain one or more
Cl atoms:
$TCl_g = 4*[CCl_4] + 3*[CHCl_3] + 2*[CH_2Cl_2] + [CH_3Cl] + 2*[Cl_2] + [HOCl] + ……$       E1
Impacts on air quality and climate are due to the high reactivity of atomic Cl produced by common
atmospheric reactions (e.g., photolysis and oxidation) of Cl-containing compounds (Riedel et al.,
2014; Sherwen et al., 2016; Haskins et al., 2018). The Cl cycle is important to atmospheric
composition in the stratosphere and troposphere, affecting species including methane, ozone, and
particles (both formation and composition), which influence air quality and climate (Solomon,
1999; Riedel et al., 2014; Young et al., 2014; Sherwen et al., 2016). High levels of some $TCl_g$
species (e.g., $Cl_2$ and carbon tetrachloride) are known to be toxic (White and Martin, 2010; Unsal
et al., 2021). The implications of many $TCl_g$ species on human health are not well understood for
low level exposure for extended periods of time. Potential health impacts of organic chlorinated
compounds include hepatotoxicity, nephrotoxicity, and genotoxicity (Unsal et al., 2021;
Henschler, 1994). Impacts of inorganic chlorinated species include the chlorination of squalene, a
major part of human skin oils, by $HOCl$ (Schwartz-Narbonne et al., 2019); respiratory irritation
and airway obstruction by $Cl_2$ (White and Martin, 2010); and increased incidence of asthma and
other chronic respiratory issues following exposure to chloramines (Massin et al., 1998).
Sources of Cl to the atmosphere are highly variable and depend on both direct emissions
and indirect regional Cl activation chemistry (Finlayson-Pitts, 1993; Raff et al., 2009; Khalil et al.,
1999). Direct emissions of $TCl_g$ can come from numerous natural and anthropogenic activities
such as, but not limited to, ocean and volcanic emissions, biomass burning, disinfection (i.e.,
household cleaning, pool emission, etc), use of solvents and heat transfer coolants, and incineration
of chlorinated wastes (Blankenship et al., 1994; Lobert et al., 1999; Keene et al., 1999; Butz et al.,
2017; Wong et al., 2017; Fernando et al., 2014). Activation of Cl is another source, occurring when
atmospheric processes transform relatively unreactive chloride ($Cl^-$, such as sea salt, NaCl) into
reactive gaseous chlorine ($Cl_y$), which will contribute to $TCl_g$. Understanding global levels of $TCl_g$
is difficult due to complex emissions and chemistry. Our best estimates come from modelling
studies combined with collaborative efforts to compose policy reports on halogenated substances,
such as the World Meteorological Organization (WMO) Scientific Assessment of Stratospheric
Ozone Depletion (WMO (World Meteorological Organization), 2018). Mixing ratio estimates of
halogenated species from this report are summed from individual measurements (e.g., National
Oceanic and Atmospheric Administration (NOAA) and Advanced Global Atmospheric Gases
Experiment (AGAGE)). The WMO report includes flask (captured gas from clean air sectors) and
in-situ measurements from field campaigns and routine sampling sites (e.g., CONvective
Transport of Active Species in the Tropics (CONTRAST)) (Prinn et al., 2018; Pan et al., 2017;
Andrews et al., 2016; Montzka et al., 2021; Adcock et al., 2018). In the most recent WMO report
(2018), a decrease of $12.7 \pm 0.9$ pptv Cl $yr^{-1}$ in total tropospheric Cl was determined for Montreal
Protocol-controlled substances (e.g., chlorofluorocarbons (CFCs) and hydrochlorofluorocarbons
(HCFCs)). The decrease in Montreal Protocol-controlled emissions has been slightly offset by an
increase in relatively short-lived substances (e.g., dichloromethane) that are not controlled by the
Montreal Protocol (WMO (World Meteorological Organization), 2018). Despite the emissions of
these regulated chlorinated species being relatively well-constrained, new sources for some of
these compounds have appeared in the recent past. For example, unexpected increases observed in
CFC-11 emissions suggested new unreported production (WMO (World Meteorological
Organization), 2018). A new source of chloroform was also recently identified and attributed to
halide containing organic matter derived from penguin excrement in the Antarctic tundra (Zhang
et al., 2021). Atmospheric levels of $TCl_g$ will additionally be impacted by emission sources that
are relatively poorly constrained, including combustion and disinfection. Increasing levels of
chlorinated species from known and unknown pathways was observed in a recent ice core study,
which estimated an increase of up to 170% of $Cl_y$ (= $BrCl + HCl + Cl + ClO + HOCl + ClNO_3 +$
$ClNO_2 + ClOO + OClO + 2 \cdot Cl_2 + 2 \cdot Cl_2O_2 + ICl$) from preindustrial times to the 1970s could be
attributed to mostly anthropogenic sources (Zhai et al., 2021).

Understanding $TCl_g$ source and sink chemistry is not only important for the ambient

atmosphere but also for indoor environments. Uncertainty in sources and levels of chemicals,
including Cl-containing compounds, indoors is related to heterogeneity in sources and individual
indoor environments, and the fact that relatively few studies have focused on indoor chemistry
compared to outdoor. The role of chlorinated species on indoor air quality has been investigated
in a few studies (Mattila et al., 2020; Wong et al., 2017; Dawe et al., 2019; Giardino and Andelman,
1996; Shepherd et al., 1996; Doucette et al., 2018; Nuckols et al., 2005). Most studies have focused
on cleaning with Cl-based cleaners, in which HOCl and other inorganic compounds have been
observed in the gas phase at high levels (Wong et al., 2017; Wang et al., 2019; Mattila et al., 2020).
Some studies have reported the presence of organic chlorinated species such as chloroform and
carbon tetrachloride above bleach cleaning solutions indoors (Odabasi, 2008; Odabasi et al., 2014),
and chloroform has been observed during water-based cleaning activities, such as showering and
clothing washing (Nuckols et al., 2005; Shepherd et al., 1996; Giardino and Andelman, 1996).

Constraining the Cl budget is critical to better understanding its contributions to climate,

air quality, and human health. Robust total Cl measurements are useful because it is not always
feasible to routinely deploy individual measurements of the large number of known Cl-containing
compounds (Table S1). As described above, estimates of $TCl_g$ from models and summed
measurements have demonstrated gaps in our knowledge. It is therefore essential to have a method
capable of measuring true $TCl_g$ to explain discrepancies between model and measured estimates
due to unknown species. Measurements of total elemental composition in the condensed phase,
including total Cl, have been used for monitoring and managing both known and unknown
compounds (Miyake et al., 2007c, a; Yeung et al., 2008; Miyake et al., 2007b; Kannan et al., 1999;
Xu et al., 2003; Kawano et al., 2007). However, $TCl_g$ methods have been limited to offline analysis
of scrubbed sample gas (e.g., flue); these methods rely on multiple extraction steps and the
application of condensed-phase total Cl analyses, such as combustion ion chromatography
(Miyake et al., 2007a; Kato et al., 2000) or neutron activation analysis (Berg et al., 1980; Xu et al.,
2006, 2007). Because offline techniques suffer from extraction uncertainties and do not have the
temporal resolution to effectively probe fast chemistry in the atmosphere, in-situ measurements of
total elemental gaseous composition have been developed for several elements (Hardy and Knarr,
1982; Veres et al., 2010; Roberts et al., 1998; Maris et al., 2003; Yang and Fleming, 2019). For
example, total nitrogen has been measured using Pt-catalyzed thermolysis coupled to online
chemiluminescence detection (Stockwell et al., 2018). Using a similar approach, we describe here
a method for $TCl_g$, where catalyzed thermolysis is coupled to a high time-resolution HCl cavity
ring-down spectrometer (CRDS). This technique relies on the complete thermolysis of $TCl_g$, which
yields chlorine atoms. These Cl atoms readily form HCl via hydrogen abstraction (R1), in this case
from propane (or its thermolysis products) that is supplied in excess.

$$Cl\ (g) + C_3H_8\ (g) \rightarrow HCl\ (g) + C_3H_7\ (g) \qquad\qquad\qquad R1$$

The objectives of this paper are to: (i) Develop and validate an instrument capable of in-

situ measurement of $TCl_g$ through conversion to HCl and detection by CRDS; and (ii) demonstrate
application of the technique to outdoor and indoor $TCl_g$ measurements.
**2. Materials and experimental methods**
**2.1.  Chemicals**

Commercially available reagents were purchased from Sigma-Aldrich (Oakville, Ontario,

Canada): dichloromethane (DCM, HPLC grade), 1-chlorobutane (CB, 99.5%), cis-1,3-
dichloropropene (DCP, 97%), trichlorobenzene (TrCB, 99%), tetrachlorobenzene (TeCB, 98%),
pentachlorobenzene (PeCB, 96 %), sodium chloride, and 52 mesh sized platinum catalyst (99.9
%). Toluene (HPLC grade) was purchased from BDH VWR (Mississauga, Ontario, Canada).
Nitrogen (grade 4.8) and propane ($C_3H_8$, 12.7% in nitrogen, v/v) gas was from Praxair (Toronto,
Ontario, Canada). Experiments used deionized water generated by a Barnstead Infinity Ultrapure
Water System (Thermo Fisher Scientific, Waltham, Massachusetts, USA; 18.2 M$\Omega$ cm$^{-1}$). A

permeation device (PD) described previously was used to generate gaseous HCl (Furlani et al., 2021). Chlorine-free zero air was generated by a custom-made zero-air generator.

## 2.2. HCl and total chlorine (HCl-TCl) instrument

The main components of the HCl-TCl (Figure 1) are platinum catalyst mesh, a quartz glass flow tube, a split-tube furnace (Protégé Compact, 1100°C max temperature, Thermcraft incorporated, North Carolina, USA), and a CRDS HCl analyzer (Picarro G2108 Hydrogen Chloride Gas Analyzer). The platinum catalyst consisted of ~2 g platinum mesh with a total combined surface area of 134 cm$^2$. Sample gas was mixed with critical orifice-regulated (Lenox laser, Glen Arm, Maryland, USA, 30 psi; SS-4-VCR-2-50) propane gas (62 ± 6 standard cubic centimetres per minute (sccm)), provided in excess prior to introduction to the furnace to promote (R1). The added propane does not fully thermolyze at temperatures < 650 °C, which can lead to spectral interferences in the CRDS analyzer (Figure S1) and should only be added when temperatures exceed 650°C (Furlani et al., 2021). All lines and fittings were made of perfluoroalkoxy (PFA) unless stated otherwise. The mixing line carrying clean air dilution flows was controlled by a 10 L min$^{-1}$ mass flow controller (MFC, GM50A, MKS instruments, Andover, Massachusetts, USA). The length of the sample gas tubing to the furnace was 0.6 m, and the transfer line between the furnace and CRDS was 0.2 m. The furnace transfer line met an overflow tee when delivering flows greater than the CRDS flowrate of 2 L min$^{-1}$. The coupled CRDS can capture transient fast HCl formation processes on the timescale of a few minutes, limited by the high adsorption activity of HCl on inlet surfaces (discussed further in Section 3.3). The CRDS collects data at 0.5 Hz. Limits of detection (LODs) for the CRDS were calculated as three times the Allan–Werle deviation in raw signal intensity when overflowing the inlet with zero air directed

into the CRDS for ~ 10 h.  The 30-sec LOD is 18 pptv and well below expected HCl from $TCl_g$
conversion (Furlani et al., 2021a).

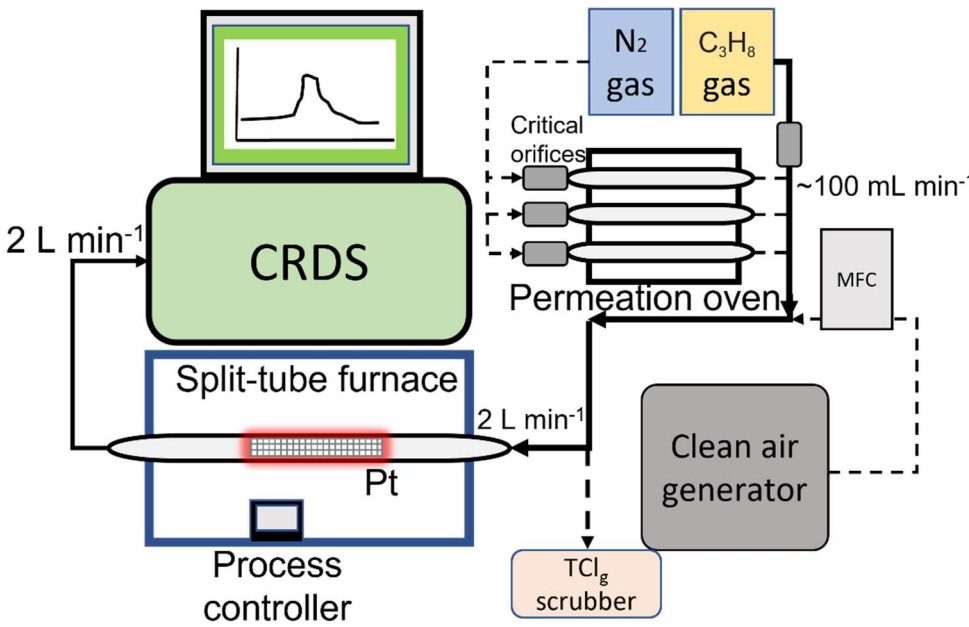


**Figure 1.** Sampling schematic showing the key components of the HCl-TCl coupled to the CRDS
analyzer. Dashed lines indicate parts of the apparatus used only during validation. Not to scale.

### 2.3.    Preparation of organochlorine permeation devices (PDs)

Organochlorine PDs were prepared as follows: approximately 200 µL of DCM, CB, or DCP

was pipetted into a 50 mm PFA tube (3 mm i.d. with 1 mm thickness), thermally sealed at one end
and plugged at the other end with porous polytetrafluoroethylene (PTFE) (13 mm length by 3.17
mm o.d.). The polymers allow a consistent mass of standard gas to permeate at a given temperature
and pressure. The method for temperature and flow control of the PDs is described in detail in Lao
et al. (2020). Briefly, an aluminum block that was temperature-controlled (Omega$^{TM}$; CN 7823,
Saint-Eustache, QC, Canada) using a cartridge heater (Omega$^{TM}$; CIR-2081/120V, Saint-Eustache,
QC, Canada) housed the PD and was regulated to $30.0 \pm 0.1$ °C. Dry $N_2$ gas flowed through a PFA
housing tube (1.27 cm o.d.) in the block that contained the PD. Stable flows of carrier gases passed
through the housing tube in the oven were achieved using a 50 µm diameter critical orifice (Lenox
laser, Glen Arm, Maryland, USA, 30 psi; SS-4-VCR-2-50) and were 120 ± 12, 99 ± 9.9 and 120
± 12 sccm for DCM, CB, and DCP, respectively. Flows were measured using a DryCal Definer
220 (Mesa Labs, Lakewood, Colorado, USA). The mass emission rate of each organochlorine from
the PDs was quantified gravimetrically over a period of approximately 4 weeks (mass accuracy ±
0.001 g). Mass emission rates for each PD were determined as 640 ± 10, 240 ± 40, and $1.20 \times 10^4$
$± 0.02 \times 10^4$ ng min$^{-1}$ (n=3, ± 1σ) at 30 °C for DCM, CB, and DCP, respectively.
**2.4. HCl-TCl optimization**

Gas phase standards of DCM, CB, and DCP were used to test the conversion efficiency of

chlorinated compounds to form HCl. Bond dissociation energies for carbon-Cl bonds typically
range between 310 and 410 kJ mol$^{-1}$ (Tables S1, S2). The split-tube furnace has a process controller
capable of increasing or decreasing temperature at a set °C min$^{-1}$, which allowed us to identify the
temperature at which enough energy was provided to break the bonds. By introducing a consistent
amount of each of the organochlorines, separately, to the HCl-TCl set over a simple temperature
ramping program we could monitor in real-time the conditions necessary to break the bonds by
measuring the formation of the resulting HCl. The operating temperature was determined when
complete conversion of the measured TCl$_g$ for the tested compounds was sustained at 100%
conversion based on PD emission rates.

To determine the optimal residence time in the quartz tube with the Pt catalyst, flows of

0.6–5.5 L min$^{-1}$ containing DCM sample gas in clean air were tested yielding a range of residence
times between 0.5 and 4.5 sec in the furnace. Temperature remained constant at 825 °C throughout
the experiment, and a dilution flow of 4.0 L min$^{-1}$ of clean air was added to the sample flow exiting
the furnace before introduction to the CRDS.

We tested the HCl transmission of the HCl-TCl at 2 mixing ratios (18 and 10 ppbv) using

a 12 M HCl PD with zero air dilution flows of 3.5 or 5 L min$^{-1}$ using a 5 L min$^{-1}$ MFC (GM50A,
MKS instruments, Andover, Massachusetts, USA). The HCl recovery through the furnace was
tested by comparing measured HCl mixing ratios through HCl-TCl to those with the furnace flow
tube replaced by a similar length of tubing. A heat gun (Master Varitemp® vt-750c) was used to
heat the flow tube entrance to ~80 °C to minimize HCl sorption. We tested the HCl-TCl conversion
efficiency for 5 different mixing ratios of three organochlorine PD standards (DCM, CB, and DCP)
under three conditions: (1) both Pt catalyst and added propane, (2) only Pt catalyst, and (3) only
added propane. Each gas was tested individually under the same conditions; sample gas from PDs
was mixed with propane and immediately diluted into clean air using a 10 L min$^{-1}$ MFC. The
dilution flows ranged from 2.2–9.0 L min$^{-1}$. The sampling lines were the same lengths as stated
previously. In this experiment, the CRDS flowrate of 2 L min$^{-1}$ was sufficient to give an optimal
residence time of 1.5 sec through the HCl-TCl (see Section 3.1). In all experiments the CRDS
subsampled through the furnace from the main transfer line and the excess gas was directed
outdoors through a waste line containing a carbon trap (Purakol, Purafil, Inc, Doraville, Georgia,
USA). We also tested the HCl-TCl conversion efficiency for two different quantities of three
chlorobenzenes (TrCB, TeCB, and PeCB). Due to their high boiling points, PDs of these
compounds could not be prepared. Instead, small volumes of approximately 1 mM solutions of
these compounds dissolved in toluene were directly introduced to the HCl-TCl while it was
sampling room air. Room air measurements of TCl$_g$ were consistently >1 ppbv. These were
measured before each experiment and did not affect the peak integration described below. With a
short piece of tubing used as an inlet, 1 and 2 µL of each compound was injected onto the inner
surface of the tubing, which was heated to ~100 °C with a heat gun to facilitate volatilization. The
resulting signals were integrated over a time period of 2.5 hours to obtain the total quantity of HCl
detected by the CRDS, which was used to calculate conversion efficiency. To account for
uncertainties in peak integration, a high and low peak area boundary was determined, with the
average peak area taken for each injection. Duplicates of each injected quantity were performed,
except for 1 μL TrCB, which was performed in triplicate.
To determine if there was any positive bias in the $TCl_g$, measurement from the conversion
of particulate chloride ($pCl^-$), NaCl aerosols were generated by flowing 2 L min$^{-1}$ of chlorine free
zero air through a nebulizer containing a solution of 2% w/w NaCl in deionized water. The aerosol
flow was then mixed with 1 L min$^{-1}$ of chlorine free dry zero air to achieve a total flow of 3 L
min$^{-1}$, The HCl-TCl (2 L min$^{-1}$) then sampled off this main mixing line. Chloride was added after
monitoring background zero air levels. After ~3 hours of measuring the converted $pCl^-$, a PTFE
filter (2 μm pore size, 47 mm diameter, TISCH scientific, North Bend, Ohio, USA) was added
inline onto the inlet of the HCl-TCl.
**2.5.  Outdoor air HCl-TCl measurements**
Outdoor air sampling was performed between 00:00 on July 7 to 20:00 on July 11, 2022
(Eastern daylight time, EDT). The sampling site was the air quality research station located on the
roof of the Petrie Science and Engineering building at York University in Toronto, Ontario,
Canada (43.7738° N, 79.5071° W, 220 m above sea level). The HCl-TCl was co-located with a
Campbell Scientific weather station paired with a cr300 datalogger. All inlet lines and fittings were
made of PFA unless stated otherwise. All indoor inlet lines and fittings were kept at room
temperature. A mass flow controller (GM50A, MKS instruments, Andover, Massachusetts, USA)
regulated a sampling flow of 14.7 L min$^{-1}$ using a diaphragm pump through a 2.4 m sampling inlet
(I.D. of 0.375") from outdoors. The outdoor air was pulled through a 2.5 μm particulate matter
cut-off URG Teflon Coated Aluminum Cyclone (URG Corporation, Chapel Hill, North Carolina,
USA) to remove larger particles and then passed through a PTFE filter (2 μm pore size, 47 mm
diameter, TISCH scientific, North Bend, Ohio, USA). The CRDS subsampled 2 L min$^{-1}$ through
the furnace off the main inlet line, yielding a total inlet flow of 16.7 L min$^{-1}$. The apparatus had
zero air overflow the inlet 1 hour prior to and after outdoor sampling. The CRDS sample flow
passed first through a PTFE filter (2 μm pore size, 47 mm diameter) and then two high efficiency
particulate air (HEPA) filters contained within the CRDS outer cavity metal compartment heat-
regulated to 45 °C. Instances of flagged instrument errors in the CRDS data during ambient
observations were removed as standard practice in quality control procedures. The dataset can be
found in Furlani et al., (2022).

### 2.6. Indoor air HCl-TCl and HOCl analyzer measurements

To test indoor applications of the HCl-TCl, a 1 m$^2$ area of laboratory floor was cleaned with
a commercial spray bottle cleaner (1.84 % sodium hypochlorite w/w) and emissions were
compared with an HOCl analyzer. The HOCl analyzer is a commercial instrument designed to
quantify gaseous hydrogen peroxide ($H_2O_2$) using CRDS (Picarro PI2114 Hydrogen Peroxide
Analyzer; Picarro Inc.). The instrument is also sensitive to HOCl due to similar absorbance
wavelengths of their first overtone stretches in the near IR. The wavelengths monitored have been
altered to selectively detect HOCl. Details on instrument calibration and validation are provided
in Stubbs et al. (2022).
The distance from the suspended 2 m inlet lines of both instruments to the floor was ~1 m.
The flowrate through the furnace and inlet was the 2 L min$^{-1}$ CRDS flowrate. The flowrate for the
HOCl analyzer was 1 L min$^{-1}$. The sectioned off area was cleaned four times, spraying 32 times
for each application using the commercial cleaner. Three of these events were measured using the
HCl-TCl and HOCl analyzer, while one event was measured using the HCl CRDS only. The
dataset can be found in Furlani et al., (2022).

## 3. Results and Discussion

### 3.1.  HCl-TCl temperature and residence time optimization

We validated this method by testing conversion efficiency of organochlorines under different
operating parameters and conditions. Testing all $TCl_g$ species is not feasible, but by testing
compounds that contain strong Cl-containing bonds, we infer at least equal efficacy of the system
in the breakage of relatively weaker Cl-containing bonds (Tables S1 and S2). We selected strong
Cl-containing bonds (i.e., alkyl, allyl, and aryl chlorides) and used them as a proxy for compounds
containing weaker Cl bonds; therefore, by demonstrating their complete conversion we set
precedent for conversion of all $TCl_g$. The temperature of the furnace is a key factor in
accomplishing complete thermolysis, and the minimum temperature of the furnace containing the
Pt catalyst to break the C-Cl bonds in DCM was determined. A simple temperature ramping
program was used to determine the breakthrough temperature. The temperature was increased at a
rate of 2.7 °C min$^{-1}$ starting at 300 °C and ending at 800 °C. The temperature breakthrough was
observed when complete conversion of the expected HCl for the tested compounds (based on PD
emission rate) was stable after reaching the optimal temperature. It was found to be ~800 °C for
the tested organochlorines (Figure S2).
Determining the optimal residence time of sample gas in the HCl-TCl is also essential for
an optimized $TCl_g$ conversion method. Using a temperature slightly above the observed
breakthrough temperature of 800 °C determined above (825 °C), six residence times were tested
with DCM, ranging from 0.5 to 4.5 seconds in the HCl-TCl (Figure 2). At each residence time the
conversion efficiency was determined, where conversion efficiency was calculated as follows:
Conversion efficiency $= \frac{\text{Measured TCl}_g}{\text{Expected TCl}_g} \times 100$ %                    E2
The optimal residence time was ~1.5 seconds, corresponding to a conversion efficiency of 100.1
± 0.1 %. The uncertainty in conversion efficiency measurements is the variability in the measured
HCl signal for 30 minutes after a signal plateau was observed. The reported uncertainty does not
include uncertainties in mixing, or turbulence induced surface effects, which we cannot quantify.
When residence times were lower (i.e., sample gas traveled more quickly through the system) than
1.5 seconds, the conversion efficiencies were lower by 2 – 10 %, the measured HCl signal was
more erratic, and it took longer to stabilize. When residence times were higher (i.e., sample gas
traveled more slowly through the system) than 1.5 seconds, the conversion efficiencies were
comparable (± 2 %), but the measured HCl suffered from longer equilibration times (~30 minutes,
more than double the 1.5 residence time) and therefore a slower response time, likely due to
increased surface effects of HCl after exiting the furnace. An optimal residence time of 1.5 seconds
was selected for all HCl-TCl experiments for its good conversion efficiency and reasonable
response time (see Table S3).

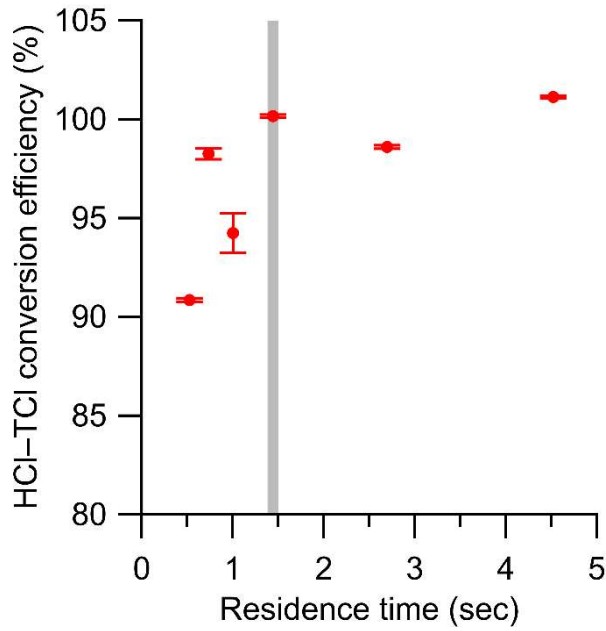


**Figure 2.** Conversion efficiency of DCM plotted against residence time in the HCl-TCl at 825 °C. Error bars represent the percent relative standard deviation of the measured HCl by the CRDS over ~30 minutes, after signal has plateaued. Grey vertical line denotes the selected residence time. Note that the error bars are represented by the precision of the instrument, and we expect there would be greater experiment-to-experiment variability.

### 3.2. HCl-TCl conversion efficiency

The efficiency of HCl throughput in the HCl-TCl was tested. Initial tests resulted in transmission efficiencies of 81.2% ± 1.4 (n = 3) and 88.1% (n = 1) for 18 ppbv and 10 ppbv HCl, respectively. At the inlet to the furnace, a small piece of the quartz tube is not heated. We hypothesized that complete transmission of HCl was hindered through sorption to that portion of quartz tube. Repeating the experiment with heat applied led to increased throughput efficiencies of 85.7% (18 ppbv, n = 1) and 93.9% (10 ppbv, n = 1). Therefore, good HCl throughput efficiency was demonstrated overall, with the cause of minor HCl losses identified to be sorption losses to room temperature glass. Conversion of particulate chloride ($pCl^-$) was observed to take place in the HCl-TCl (Figure S3), but once a filter was introduced the signal returned to background levels. Thus, to capture only gaseous $TCl_g$ from samples that may contain particulate chloride, a

particulate filter should be used. Use of a filter could introduce blow on (i.e., partitioning of semi-
volatile species) and/or blow off (i.e., processing of particulate chloride) artifacts. We have
previously shown that HCl—likely to be the most surface-active component of $TCl_g$—is not
greatly impacted by the presence of filters (Furlani et al., 2021), indicating blow on effects are
likely minimal. However, the extent to which blow on effects should be considered will depend
on the composition of the $TCl_g$ mixture and the temperature. Blow off effects will depend on
ambient particulate chloride levels and can be mitigated by regularly changing the filter to prevent
significant particulate chloride accumulation.
The conversion efficiency of each of the two alkyl chlorine and one allyl chlorine compounds
using the HCl-TCl was tested at 5 different mixing ratios. See Table S4 for summary of mixing
ratios used; all lower mixing ratios were generated by diluting the highest mixing ratio of each
compound by chlorine-free zero air. All three showed good linearity and near 1:1 correlation with
the HCl expected to be formed from the PD under standard operating conditions (Figure 3). Due
to differences in PD emission rates, the values in Figure 3 are normalized to the highest mixing
ratio to visualize comparisons more easily. With both Pt and propane the HCl-TCl conversion was
$99.6 \pm 3.2$, $104.8 \pm 5.6$, and $102.7 \pm 7.8\%$ for DCM, CB, and DCP, respectively (Table 1), as the
average conversion efficiency ± relative standard deviation. From Figure 3 the comparison
between expected and measured $TCl_g$ is illustrated by near unity in the orthogonal distance
regression slope (±1σ, the error in the regression analysis), and was $0.996 \pm 0.012$, $1.048 \pm 0.060$,
and $1.027 \pm 0.061$ for DCM, CB, and DCP, respectively. With only the Pt catalyst, the HCl-TCl
conversion was $80.7 \pm 0.4$, $54.1 \pm 1.6$, and $54.3 \pm 3.5\%$ for DCM, CB, and DCP, respectively
(Figure S4, Table 1). This result indicates the added hydrogen source (propane) is needed to
promote R1. Although necessary in this laboratory scenario, some ambient conditions may be rich
enough in hydrogen-containing molecules that excess propane is not needed. However, providing
propane in excess ensures the presence of an abundance of hydrogen atoms that can be readily
abstracted by Cl atoms via R1. When the Pt catalyst was removed, the HCl-TCl conversion was
$94.4 \pm 4.6$, $44.2 \pm 0.9$, and $41.7 \pm 3.4\%$ for DCM, CB, and DCP, respectively (Figure S4, Table
1). The observed dependence of the Pt catalyst indicates that a reactive surface is important

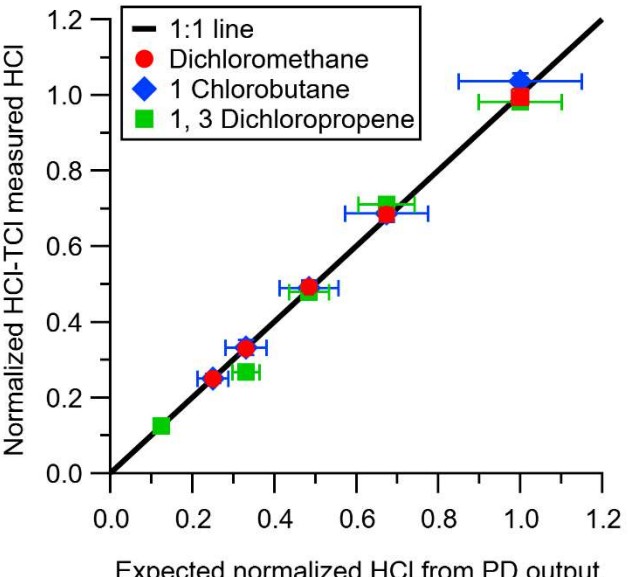


**Figure 3.** HCl measured by CRDS plotted against the expected HCl from HCl-TCl converted
DCM (red circle), 1-chlorobutane (blue diamond), and 1,3-dichloropropene (green square) under
condition (1). All values are normalized to the highest expected HCl concentration to better
illustrate deviations from unity (black line). Error bars on the y-axis represent $1\sigma$ in the HCl signal
over 10 minutes. Error bars on the x-axis represent the uncertainty in the PD used to generate
DCM.
**Table 1.** Conversion efficiency for tested Cl-containing compounds under different conditions
(both Pt and propane; Pt only; propane only). Note that chlorobenzenes were only tested under Pt
and propane conditions.

| Tested TCl$_g$ species | Cl bond dissociation energy (kJ mol$^{-1}$) | Conversion efficiency (%) | | |
|---|---|---|---|---|
| | | Pt and propane | Pt only | Propane only |
| Dichloromethane (DCM)[a] | 310 | $99.6 \pm 3.2$ | $80.7 \pm 2.4$ | $94.4 \pm 6.6$ |
| 1-Chlorobutane (CB)[a] | 410 | $104.8 \pm 5.6$ | $54.1 \pm 6.6$ | $44.2 \pm 5.9$ |
| 1, 3-Dichloropropene (DCP)[a] | 350 | $102.7 \pm 7.8$ | $54.3 \pm 5.2$ | $41.7 \pm 5.1$ |

| | | | | |
|---|---|---|---|---|
| Trichlorobenzene (TrCB)[b] | 400 | 97.0 ± 19.9 | | |
| Tetrachlorobenzene (TeCB)[b] | 400 | 90.6 ± 10.3 | | |
| Pentachlorobenzene (PeCB)[b] | 400 | 90.2 ± 14.8 | | |

[a]Conversion efficiency was determined from the orthogonal distance regression slope and ± σ and propagated error from individual permeation devices.

[b]Conversion efficiency was determined directly by the quantity (mol) of HCl measured from liquid injections of 1 mM standards. The error represents ±σ of measurements for n = 5 (TrCB) or n = 4 (TeCB, PeCB) injections.

to achieve complete thermolysis at 825 °C. The relatively higher conversion for DCM in the absence of the Pt catalyst or hydrogen source may be attributed to its lower bond dissociation energy (310 kJ mol$^{-1}$) compared to estimated bond dissociation energies for CB and DCP (CB inferred from Table S2 (~410 kJ mol$^{-1}$), and DCP from tetrachloroethylene (350 kJ mol$^{-1}$ in Table S1)). It is possible that a higher temperature could lead to full conversion of TCl$_g$ in the absence of Pt catalyst; however, that was not explored in this study. To further validate the HCl-TCl, the conversion efficiency of three aryl chlorine compounds were tested under the final operating conditions (i.e., in the presence of both Pt and added propane). The TCl$_g$ measured from the three aryl compounds was unity, within the uncertainty of the measurement (Table 1).

The results for all six compounds show that the HCl-TCl is capable of complete conversion of mono and polychlorinated species on sp$^3$ and sp$^2$ carbons using the determined temperature and flow conditions. The complete thermolysis of the strongest C-Cl bond on the primary alkyl chloride (CB) demonstrates the efficacy of the HCl-TCl. Breaking these relatively strong C-Cl bonds, with consistent conversion efficiency across alkyl, allyl, and aryl C-Cl bonds, is a good proof of concept for complete conversion of all bonds of similar or weaker bond energies that characterize all other TCl$_g$. To practically validate the HCl-TCl under real-world conditions with atmospherically relevant TCl$_g$ mixtures and mixing ratios we also deployed and configured the system to measure outdoor and indoor air.

### 3.3. Performance metrics of HCl-TCl

Using a flow of zero air through the HCl-TCl, method limits of detection (LODs) were calculated as three times the Allan-Werle deviation (Figure 4) when overflowing a 20 cm inlet (3.17 mm i.d.) with zero air for one hour. The LODs determined in the measurements for 2 second, 1 minute, 5 minute, and 1 hour integration times were 73, 15, 10, and 8 pptv, respectively. The response time of the instrument was assessed during experiments with DCM, CB, and CP. The time for the signal to decay after removal of the PDs was determined to 37 % (1/e) and 90 % ($t_{90}$) of the maximum signal. The maximum time to achieve 1/e was 23 seconds, while the maximum time to achieve $t_{90}$ was 189 seconds (Table S3). These are comparable to the response times for the HCl CRDS instrument itself (Furlani et al., 2021), suggesting the addition of the inlet furnace has a modest impact on the residence time. Given the high mixing ratios used to test the response times, we argue that under most conditions relevant to indoor and outdoor atmospheric chemistry, a sample integration time of one minute will minimize any time response effects. Data for outdoor and indoor sampling described in Sections 3.4 and 3.5 were therefore averaged to one minute. During all experiments with gaseous reagents, no evidence of catalyst performance degradation was observed.

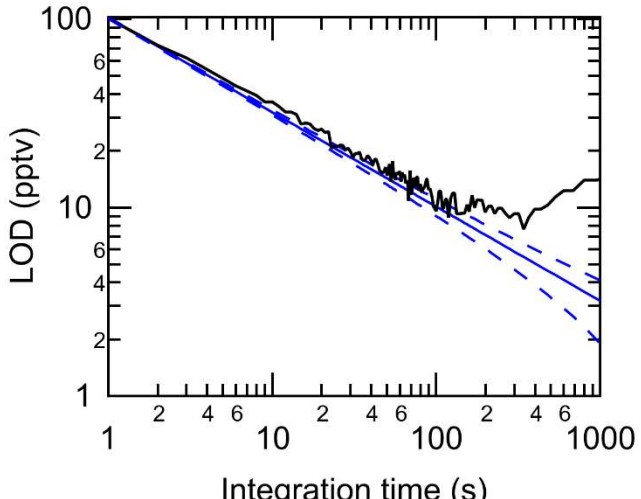

**Figure 4.** Allan-Werle deviation (3σ) in the HCl-TCl purged with zero-air (black line) shown with the ideal deviation (no drift, solid blue line) and associated error in the deviation (dashed blue line).

### 3.4. HCl-TCl applications to outdoor air

We deployed the system to measure ambient outdoor air, which we compare to the expected $TCl_g$ range from complete thermolysis of previously measured Cl-containing compounds, estimated to be between 3.3 and 19 ppbv (Table S1). Global background levels of long-lived chlorine-containing species ($LLCl_g$) are well established (WMO (World Meteorological Organization), 2018) and were calculated by equation 3 using data from Table S1:

$LLCl_g = 3*[CCl_3F] + 2*[CCl_2F_2] + 4*[CCl_2FCCl_2F] + 4*[CCl_3CClF_2] + 3*[CCl_3CF_3] + 2*[CClF_2CClF_2] + 2*[CCl_2FCF_3] + [CClF_2CF_3] + [CHClF_2] + [CH_2ClCF_3] + 2*[CH_3CCl_2F] + [CBrClF_2] + 4*[CCl_4]$        E3

A global background for $LLCl_g$ of approximately 2.6 ppbv is expected ((WMO (World Meteorological Organization), 2018), Table S1). The maximum, minimum, and median of observed ambient $TCl_g$ were 536.3, 2.0, and 3.1 ppbv, respectively (Figure 5). Measurements of HCl alone were not made during these periods but reported ranges of HCl mixing ratios for this sampling location from Furlani et al. (2021) and Angelucci et al. (2021) were typically below 110 pptv, with intermittent events up to 600 pptv. The filter present in the inlet was unlikely to have led to artifacts in this measurement. Particulate chloride is negligible in continental summertime environments (Kolesar et al., 2018), indicating blow off artifacts would be minimal. Most ambient $TCl_g$ measurements were above the expected mixing ratio of $LLCl_g$. It is possible that semi-volatile chlorinated species could have partitioned to the filter, acting as a blow on effect, and leading to an underestimate of $TCl_g$. However, the warm temperatures during sampling (13 to 31 °C) and high observed $TCl_g$ levels suggest this was not a large effect. There is clear evidence of $TCl_g$ sources beyond $LLCl_g$ at the sampling site, with several plumes of elevated $TCl_g$ intercepted. For example, the maximum $TCl_g$ measurement (536.3 ppbv) was made in a plume just after noon on

July 7. Another plume was detected on July 11, with a maximum TCl$_g$ of 42.1 ppbv. Though the
purpose of this study was not to determine sources of TCl$_g$, we observed that plumes containing
elevated TCl$_g$ arrived from the S-SW of the sampling site, where several facilities that had reported
tens to thousands of kg of yearly emissions to air of Cl-containing species are located (Figure S5).

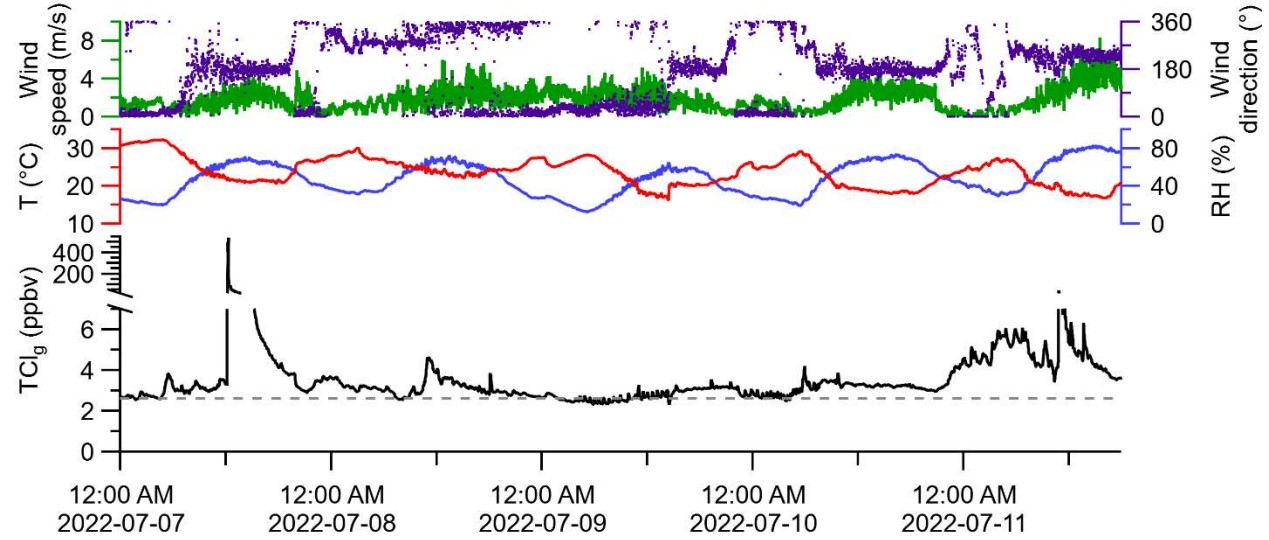


**Figure 5.** Monitoring meteorological conditions and one-minute averaged TCl$_g$ in outdoor air
through HCl-TCl from July 7 to 11, 2022. Grey dashed line represents the background mixing
ratio for LLCl$_g$.
**3.5. HCl-TCl application to indoor cleaning**

We applied a chlorine-based cleaning product four times in a well-lit indoor room and

measured TCl$_g$ using the HCl-TCl and HOCl analyzer during three of the cleaning events (Figure
6). One cleaning experiment was done without the HCl-TCl and had a maximum of 370 pptv HCl.
These levels are comparable to peak HCl levels of ~500 pptv observed from surface application
of bleach (Dawe et al., 2019). Consistent with previous speciated measurements (Mattila et al.,
2020; Wong et al., 2017), HCl, HOCl, and TCl$_g$ levels increased rapidly over ~5 minutes after the
application of the cleaning product. The maximum levels of TCl$_g$ from HCl-TCl during application
1, 2, and 3, were 49.2, 80.0, and 69.7 ppbv, respectively. The maximum levels of HOCl from
applications 1, 2, and 3, were 19.6, 24.2, and 16.8 ppbv, respectively, corresponding to 24 to 40 %
of peak $TCl_g$ and 14 to 22 % of integrated $TCl_g$. These $TCl_g$ levels were several times higher than
most observed in outdoor air (Section 3.4) and were within the range expected from previous
experiments (Table S1). The levels of chlorinated species observed during bleaching events is
variable, between 15 to 100s of ppbv (Mattila et al., 2020; Odabasi, 2008; Wang et al., 2019; Wong
et al., 2017). By comparison, our highest observed mixing ratio was 80 ppbv. Because the
multiphase chemical processes involved in bleach application are complex and poorly understood,
it is difficult to compare levels between similar studies, given that the underlying ambient
conditions can be very different. In addition, physical parameters, such as volume of cleaning
solution applied, room size, and ventilation, can all affect observed mixing ratios. For example,
studies have observed that gaseous $NH_3$ partitioning into aqueous bleach can produce large and
variable amounts of chloramines, $NH_2Cl$, $NHCl_2$, and $NCl_3$ (Mattila et al., 2020; Wong et al.,
2017). In our experiments, there was on average $82 \pm 4$ % of integrated $TCl_g$ that could not be
accounted for by the HOCl measurement. Additional chlorinated species that have previously been
observed to be emitted from surface bleaching include $ClNO_2$, $NH_2Cl$, $NHCl_2$, $NCl_3$, and several
chlorinated organics (Odabasi, 2008; Mattila et al., 2020; Wong et al., 2017) which likely also
contributed to our measured $TCl_g$. We observed that $TCl_g$ decayed ~15% faster than the air
exchange rate (0.72 $h^{-1}$), indicating additional chemical loss pathways or surface interactions
(Figure S6). We observed a shorter lifetime of HOCl relative to $TCl_g$, which is consistent with
faster decay rates observed for HOCl and similar $TCl_g$ species by Wong et. al., (2017). The HOCl
started decreasing after ~300 s had elapsed while the $TCl_g$ levels were still increasing. This
suggests that reactions involving HOCl may have led to additional $TCl_g$ species, which has been
observed in laboratory studies (Wang et al., 2019).
In-situ measurements of $TCl_g$ could provide additional insight into sources of chlorinated
species to indoor environments by creating a total inventory from which the contributions of
individual measured species can be compared and used to elucidate unknown $TCl_g$ levels and
mechanisms in real-time. Furthermore, several chlorinated species that have previously been
observed to be emitted from surface bleaching, including $Cl_2$, $HOCl$, $ClNO_2$, $NH_2Cl$, $NHCl_2$, and
$NCl_3$ (Mattila et al., 2020; Wong et al., 2017), have been measured by chemical ionization mass
spectrometry (CIMS). Quantifying chlorinated species using CIMS remains challenging due to the
required calibrations and difficulty in generating pure gas phase standards. It is therefore desirable
to have a technique such as the one proposed in this study that does not require calibrations or
knowledge of potential unknown $TCl_g$ species. A combination of the two methods would help
constrain the total levels while still observing speciation for key $TCl_g$ species.

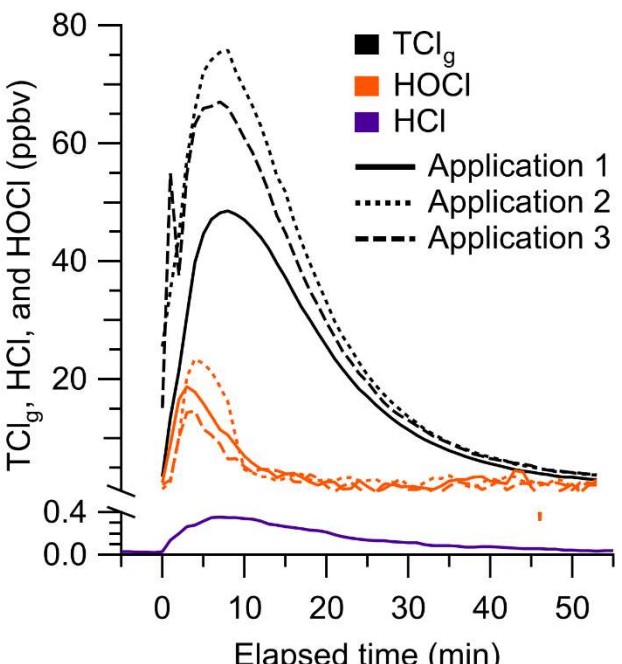


**Figure 6.** One-minute average HCl (purple), HOCl (orange), and $TCl_g$ (black) observed during
cleaning spray events. Mixing ratios were background corrected prior to each cleaning event. Each
subsequent application of cleaner is illustrated by a lighter shade for HOCl and $TCl_g$.

## 4. Conclusions

**4. Conclusions**
In this work we developed, optimized, validated, and applied a method capable of converting
$TCl_g$ into gaseous HCl for detection by CRDS. Our $TCl_g$ measurement technique, the HCl-TCl, is
composed of a platinum catalyst mesh inside a quartz glass flow tube all contained within a split-
tube furnace. The temperature and flow rate were optimized at 825 °C and 1.5 seconds,
respectively using DCM. These conditions were validated by the complete conversion of
organochlorine compounds with strong C-Cl bonds. The HCl-TCl was used to measure $TCl_g$
outdoors, observing a range of 2.0 to 536.3 ppbv. Levels mostly exceeded the expected background
mixing ratio of $LLCl_g$. We also applied the HCl-TCl to an indoor environment during commercial
bleach spray cleaning events and observed varying increases in $TCl_g$ (50–80 ppbv), which was in
reasonable agreement with levels observed in previous speciated measurements. The agreement of
HCl-TCl outdoor and indoor measurements with available bottom-up estimates indicates its
efficacy under real-world scenarios. Rapid changes in $TCl_g$ were observed in both outdoor and
indoor environments indicating the utility of an in-situ technique to constrain the sources and
chemistry of $TCl_g$, as well as its impact on air quality, climate, and health. We anticipate this
approach could be used in several applications, including comparisons to speciated measurements
of chlorinated compounds and to further explore Cl reactivity and cycling with respect for indoor
and outdoor $TCl_g$.
**Acknowledgements**
We acknowledge the Sloan Foundation and Natural Sciences Engineering and Research Council
of Canada for funding. We thank Melodie Lao and Yashar Iranpour for collecting air exchange
rate data, Andrea Angelucci for collecting meteorological data, Dirk Verdoold for the custom
quartz tube, and Chris Caputo, John Liggio, Rob McLaren, and Trevor VandenBoer for helpful
discussions. PME thanks the European Research Council. TFK is a Canada Research Chair in
Environmental Analytical Chemistry. This work was undertaken, in part, thanks to funding from
the Canada Research Chairs program.
**Author contributions**
TCF, RY, JS, and LRC collected and analyzed the data. TCF, RY, LRC, and CJY conceived of
and designed the experiments with input from PME and TFK. Funding was obtained by TFK and
CJY. The manuscript was written by TCF, RY, and CJY with input from all authors.
**Data availability**
Outdoor and indoor datasets can be found in Furlani et al. (2022,
https://doi.org/10.20383/103.0649).
**Competing interests**
The authors declare no competing interests.

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
