# Peer review of "Development and Validation of a New In-Situ Technique to Measure Total Gaseous Chlorine in Air"

_Atmospheric Measurement Techniques, 2022_

## Author Comment (AC1)

We thank the Reviewer for their detailed and constructive comments. Please find our responses below in blue highlight, with changes to the manuscript indicated in **bold**.

Furlani et al. describe a new instrument for quantification of "total gaseous chlorine" in air. The instrument uses an inlet furnace to which propane and a Platinum catalyst are added to quantify the generated hydrochloric acid by a commercial cavity ring-down spectrometer. The instrument is conceptually similar to existing "total element" measurement techniques such as Cy (Roberts et al., 1998) and Nt (Stockwell et al., 2018). Its performance was evaluated using diffusion sources containing dichloromethane, 1-chlorobutane or cis-1,3-dichloropropene. Sample data of outdoor air and indoor air measurements are presented. The indoor air data were supplemented by HOCl measurements.

I have major concerns with this manuscript which, in my opinion, requires more data to verify the instrument's performance.

We thank the Reviewer for their detailed comments on the manuscript. We have undertaken several additional experiments to further validate the method. The experiments are detailed in response to specific comments below.

Major comments

(1) The list of Cl containing compounds (Table S1) is incomplete and omits, amongst others, cyanogen and acid halides and chlorinated aromatic compounds such as chlorinated dioxins and other legacy organochlorine pesticides. The list of compounds should be as complete as possible for this to be a "total" measurement. I would also suggest not burying this list in the SI but to move it in to the main manuscript.

We agree with the Reviewer that we have not included all possible Cl-containing compounds. We have focused on the compounds most accurately measured and/or most likely to be present in the atmosphere, because we believe the quality and applicability of the measurement is not dependent on how well we know the current identity and level of chlorine-containing compounds in the atmosphere. For example, to our knowledge, atmospheric levels of chlorinated dioxins and chlorinated pesticides are not well known. There are likely to be hundreds of individual chlorinated species in the atmosphere, many of which have not yet been identified that would each be present at low levels (e.g., Fernando et al., (2014)). We have clarified the caption of Table S1:

> "**Table S1**. Summary of tropospheric mixing ratios, bond dissociation energies, and atmospheric lifetimes for **Cl-containing species that are frequently measured and/or expected to be present in appreciable amounts in the troposphere.**"

(2) Cly is defined on line 86 but a clear definition (with stoichiometric factors) of the various compounds that make up total gaseous chlorine TClg was absent (something like TClg = HCl + CH3Cl + 2CH2Cl2 + 3CHCl3 + 4CCl4 ...). This omission is odd considering TClg is the new quantity supposedly being measured.

Unfortunately, we are not able to fully define $TCl_g$. One of our major motivators for developing this technique was the ability to assess what fraction of the total measured chlorine cannot be

accounted by current measurements. However, we can provide a definition that, like other concepts in atmospheric chemistry (e.g., OH reactivity) acknowledges the unknown unknowns inherent to the term. This has been added to the introduction:

"In this work, total gaseous Cl ($TCl_g$) refers to **all gas-phase Cl-containing species weighted to their Cl content**, including both inorganic and organic species. **While groups of chlorinated species are often considered based on reactivity considerations (e.g., reactive chlorine, $Cl_y$), $TCl_g$ includes all molecules that contain one or more Cl atoms:**

$$TCl_g = 4*[CCl_4] + 3*[CHCl_3] + 2*[CH_2Cl_2] + [CH_3Cl] + 2*[Cl_2] + [HOCl] + \ldots \ldots \quad E1"$$

We can also explicitly define the known long-lived gaseous chlorine ($LLCl_g$). This represents the sum of chlorine-containing chemicals that are routinely measured and have long enough lifetimes that their atmospheric levels are expected to be relatively well-mixed in the atmosphere. We have introduced this concept into the manuscript:

"We deployed the system to measure ambient outdoor air, which we compare to the expected $TCl_g$ range from complete thermolysis of **previously measured Cl-containing compounds**, expected to be between 3.3 and 19 ppbv (Table S1). **Global background levels of long-lived chlorine-containing species ($LLCl_g$) are well established** (WMO (World Meteorological Organization), 2018) **and were calculated by equation 3 using data from Table S1:**

$$LLCl_g = 3*[CCl_3F] + 2*[CCl_2F_2] + 4*[CCl_2FCCl_2F] + 4*[CCl_3CClF_2] + 3*[CCl_3CF_3] + 2*[CClF_2CClF_2] + 2*[CCl_2FCF_3] + [CClF_2CF_3] + [CHClF_2] + [CH_2ClCF_3] + 2*[CH_3CCl_2F] + [CBrClF_2] + 4*[CCl_4] \quad E3$$

A global background **for $LLCl_g$** of approximately 2.6 ppbv is expected ((WMO (World Meteorological Organization), 2018), Table S1)."

Further along the same vein, the concept of total chlorine exists since stratospheric times (Cly), and many in the community use terms such as "active chlorine" or "reactive chlorine" (Zhai et al., 2021). The manuscript needs more discussion on where this new measurement of TClg fits in the broader picture.

While groups of chlorinated chemicals have been thought of together for many years, these definitions differ from $TCl_g$. The example the Reviewer provides of $Cl_y$ is defined as reactive chlorine, for which the stratospheric definition is (Seinfeld and Pandis, 2006):

$$Cl_y = Cl + ClO + HOCl + ClONO_2 + HCl$$

The same textbook defines $CCl_y$ as the chlorine contained in a few carbon and chlorine-containing molecules. Regardless, there has not yet been a definition of $TCl_g$ as we have defined it here. We also consider the chlorine-containing molecules weighted by their chlorine content. We have added text to the introduction to clarify this point:

"In this work, total gaseous Cl ($TCl_g$) refers to all gas-phase Cl-containing species **weighted to their Cl content**, including both inorganic and organic species. **While groups of chlorinated**

**species are often considered based on reactivity considerations (e.g., reactive chlorine, $Cl_y$), $TCl_g$ includes all molecules that contain one or more Cl atoms.”**

(3) The authors argue that, because the bond dissociation energy is higher in HCl than all other compounds listed in Table S1, it suffices to only evaluate the conversion efficiency of 3 compounds and assume that the remainder will fall in line and fully convert to HCl when sampled. While this may be true from a thermodynamic point of view, kinetic barriers to dissociation may come into play because the instrument is operated with a relatively short residence time of ~1.5 s. More experiments should be conducted to verify the instrument's response to others chlorine containing gases (for example, carbon tetrachloride, hexachlorobenzene, and perhaps a chlorinated dioxin).

We have added conversion efficiency tests to three aryl organochlorine compounds. Our conversion efficiency tests now encompass alkyl, allyl, and aryl organochlorines, with six separate compounds tested. The text and Table 1 have been updated to reflect these additional experiments:

Section 2.4:

**“We tested the HCl-TCl conversion efficiency for two different quantities of three chlorobenzenes (TrCB, TeCB, and PeCB). Due to their high boiling points, PDs of these compounds could not be prepared. Instead, small volumes of approximately 1 mM solutions of these compounds dissolved in toluene were directly introduced to the HCl-TCl while it was sampling room air. Room air measurements of HCl were consistently <100 pptv, therefore should not influence experimental measurements. With a short piece of tubing used as an inlet, 1 and 2 µL of each compound was injected onto the inner surface of the tubing, which was heated to ~100 °C with a heat gun to facilitate volatilization. The resulting signals were integrated over a time period of 2.5 hours to obtain the total quantity of HCl detected by the CRDS, which was used to calculate conversion efficiency. To account for uncertainties in peak integration, a high and low peak area boundary was determined, with the average peak area taken for each injection. Duplicates of each injected quantity were performed, except for 1 µL TrCB, which was performed in triplicate.”**

Section 3.2:

**“To further validate the HCl-TCl, the conversion efficiency of three aryl chlorine compounds were tested under the final operating conditions (i.e., Condition 1, in the presence of both Pt and added propane). The $TCl_g$ measured from the three aryl compounds was unity, within the uncertainty of the measurement (Table 1).**

The results for all **six** compounds show that the HCl-TCl is capable of complete conversion of mono and polychlorinated species on $sp^3$ and $sp^2$ carbons using the determined temperature and flow conditions. The complete thermolysis of the strong**est** C-Cl bond on the primary alkyl chloride (CB) demonstrates the efficacy of the HCl-TCl. Breaking these relatively strong C-Cl bonds, **with consistent conversion efficiency across alkyl, allyl, and aryl C-Cl bonds**, is a good proof of concept for complete conversion of all bonds of similar or weaker bond energies that characterize all other $TCl_g$.”

**Table 1.** Conversion efficiency for tested Cl-containing compounds under different conditions (both Pt and propane; Pt only; propane only).

| Tested TCl$_g$ species | Cl bond dissociation energy (kJ mol$^{-1}$) | Conversion efficiency (%) | | |
|---|---|---|---|---|
| | | **Pt and propane** | **Pt only** | **Propane only** |
| Dichloromethane (DCM)[a] | 310 | 99.6 ± 3.2 | 80.7 ± 2.4 | 94.4 ± 6.6 |
| 1-Chlorobutane (CB)[a] | 410 | 104.8 ± 5.6 | 54.1 ± 6.6 | 44.2 ± 5.9 |
| 1, 3-Dichloropropene (DCP)[a] | 350 | 102.7 ± 7.8 | 54.3 ± 5.2 | 41.7 ± 5.1 |
| **Trichlorobenzene (TrCB)[b]** | **400** | **97.0 ± 19.9** | | |
| **Tetrachlorobenzene (TeCB)[b]** | **400** | **90.6 ± 10.3** | | |
| **Pentachlorobenzene (PeCB)[b]** | **400** | **90.2 ± 14.8** | | |

[a]Conversion efficiency was determined from the orthogonal distance regression slope and ± σ and propagated error from individual permeation devices.
[b]Conversion efficiency was determined directly by the quantity (mol) of HCl measured from liquid injections of 1 mM standards. The error represents ±σ of measurements for n = 5 (TrCB) or n = 4 (TeCB, PeCB) injections.

(4) The instrument was "validated" using diffusion sources that are calibrated only gravimetrically and are thus not necessarily accurate. Many of the compounds listed in Table S1 can be quantified by established instrumental methods such as GCMS. The output of the diffusion sources could (and probably should) have been calibrated by another instrumental method such as GCMS, and the response compared to that of the new instrument described here.

We respectfully disagree with the Reviewer. Gravimetric analysis has long been an accepted technique for quantifying the output of permeation tubes made with pure substances (e.g., Scaringelli et al., (1970), Mitchell (2000)). If one were to purchase a validated permeation tube from at least one typically used vendor, their permeation rates are determined gravimetrically (e.g., https://www.vici.com/calib/calib.php).

(5) Interferences need to be evaluated - for example, would the instrument respond to sea salt aerosol or aerosol containing non-volatile Cl-containing compounds or ammonium chloride by evaporating HCl in the oven? Is the conversion efficiency reduced when flame retardants such as PDBEs (present in indoor air) are sampled? Spectral interference from propane was mentioned - are there other spectral interferences?

We thank the Reviewer for comments on the potential sea salt bias and have taken steps to address these concerns. We have tested the impacts of particulate chloride and added Figure S3 to the SI. As seen in Fig S3, we do see some conversion of chloride. Though we have chosen not to quantify it here because it is outside the desired scope of our measurement, it can be added to the list of supplemental applications. The addition of a 2 micron filter returned everything to background levels.

Text has been added to the methods (Section 2.4):

**"To determine if there was any positive bias in the TCl$_g$ measurement from the conversion of particulate chloride (pCl$^-$), NaCl aerosols were generated by flowing 2 L min$^{-1}$ of chlorine free zero air through a nebulizer containing a solution of 2% w/w NaCl in**

**deionized water. The aerosol flow was then mixed with 1 L min⁻¹ of chlorine free dry zero air to achieve a total flow of 3 L min⁻¹, The HCl-TCl (2 L min⁻¹) then sampled off this main mixing line. Chloride was added after monitoring background zero air levels. After ~3 hours of measuring the converted pCl⁻, a PTFE filter (2 μm pore size, 47 mm diameter, TISCH scientific, North Bend, Ohio, USA) was added inline onto the inlet of the HCl-TCl.”**

Text has been added to the results and discussion (Section 3.1):

**“Conversion of particulate chloride (pCl⁻) was observed to take place in the HCl-TCl (Figure S3), but once a filter was introduced the signal returned to background levels. Thus, to capture only gaseous TClg from samples that may contain particulate chloride, a particulate filter should be used.”**

[Figure]

**“Figure S3. Testing the impacts of added particulate chloride (pCl⁻)  HCl-TCl  TClg was monitored _under a flow of Cl-free air,_ while pCl⁻  added _to the air stream_ (red vertical dashed line and  a Teflon filter  was _placed in front of the HCl-TCl inlet_  (blue vertical dashed line) . _Adding the_ filter showed complete reduction of the pCl⁻ signal.”**

Concentrations of gas phase flame retardant compounds would be too low to impact the scope of our studies; the excess of propane added in our experimental setup will offset any radical impeding properties of flame retardants.

Volatile organics can interfere with the spectral fitting of the HCl by this instrument. This was previously described in detail in the paper describing this measurement (Furlani et al., 2021), which is cited in the text in the discussion of interferences.

Was the instrument's response ever evaluated by adding a known amount of DCM, CB and DCP while sampling ambient air to verify the absence of matrix effects?

Unfortunately, due to the unknown ambient concentrations of $TCl_g$ we are not able to decouple the two measurements and access matrix effects. Also, producing low level ~3 ppbv mixing ratios comparable to outdoor air of our tested PDs is currently not possible.

(6) Was the transmission efficiency of HCl through the oven at normal operating conditions (and in the absence and presence of Pt and propane ever evaluated)? If so, these data should be included in this manuscript.

Details of HCl transmission experiments have now been added to the manuscript.

Section 2.4:

**"We tested the HCl transmission of the HCl-TCl at 2 mixing ratios (18 and 10 ppbv) using a 12 M HCl PD with zero air dilution flows of 3.5 or 5 L min$^{-1}$ using a 5 L min$^{-1}$ MFC (GM50A, MKS instruments, Andover, Massachusetts, USA). The HCl recovery through the furnace was tested by comparing measured HCl mixing ratios through HCl-TCl to those with the furnace flow tube replaced by a similar length of tubing. A heat gun (Master Varitemp® vt-750c) was used to heat the flow tube entrance to ~80 °C to minimize HCl sorption."**

Section 3.2:

**"The efficiency of HCl throughput in the HCl-TCl was tested. Initial tests resulted in transmission efficiencies of 81.2% (n = 3) and 88.1% (n = 1) for 18 ppbv and 10 ppbv HCl, respectively. At the inlet to the furnace, a small piece of the quartz tube is not heated. We hypothesized that complete transmission of HCl was hindered through sorption to that glass. Repeating the experiment with heat applied led to increased throughput efficiencies of 85.7% (18 ppbv, n = 1) and 93.9% (10 ppbv, n = 1). Therefore, good HCl throughput efficiency was demonstrated overall, with the cause of minor HCl losses identified to be sorption losses to room temperature glass."**

(7) The major components of TClg , according to Table S1, are HCl and ClNO2. Have the authors considered implementing a switch to alternate between HCl and TClg ? What happens when the oven temperature is scanned while sampling air? Are plateaus observed as in the method developed by the Cohen group for NOy?

While previous measurements suggest that HCl and $ClNO_2$ should be the major inorganic components of $TCl_g$, we expect that levels of organic compounds should be present in higher quantities under most conditions.

We did not switch to sample between HCl and $TCl_g$. Unlike with sampling $NO_x$, switching between very different quantities of surface-active HCl (such as those that would be obtained for $TCl_g$ compared to ambient HCl) is non-trivial. The response time between these values, often different by orders of magnitude, would be long enough that it would make switching between these two measurements difficult (see Furlani et al. 2021).

We did not perform an experiment at different temperatures for ambient data. Because ambient levels of TCl$_g$ are not constant, this would effectively be changing two variables at once, which we did not believe would be informative.

(8) More instrumental figures of merit should be provided. For example, an LOD is provided for the CRDS, but not for TClg (which is likely worse). What is the rise time of this instrument to sudden concentration changes (how quickly does the instrument response return to zero when chlorine-free air is added to the inlet)? What is the instrument's linear dynamic range? Does the catalyst age or need to be changed over time?

We have added a section to the manuscript called "**Performance metrics of the HCl-TCl**" (Section 3.3) that includes information on the limit of detection and response time of the method.

**"3.3 Performance metrics of HCl-TCl**

**Using a flow of zero air through the HCl-TCl, method limits of detection (LODs) were calculated as three times the Allan-Werle deviation (Figure 4) when overflowing a 20 cm inlet (3.17 mm i.d.) with zero air for one hour. The LODs determined in the CRDS measurements for 2 second, 1 minute, 5 minute, and 1 hour integration times were 73, 15, 10, and 8 pptv, respectively. The response time of the instrument was assessed during experiments with DCM, CB, and CP. The time for the signal to decay after removal of the PDs was determined to 37 % (1/e) and 90 % (t$_{90}$) of the maximum signal. The maximum time to achieve 1/e was 23 seconds, while the maximum time to achieve t$_{90}$ was 189 seconds (Table S3). These are comparable to the response times for the HCl CRDS instrument itself** (Furlani et al., 2021)**, suggesting the addition of the inlet furnace has a modest impact on the residence time. Given the high mixing ratios used to test the response times, we argue that under most conditions relevant to indoor and outdoor atmospheric chemistry, a sample integration time of one minute will minimize any time response effects. Data for outdoor and indoor sampling described in Sections 3.4 and 3.5 were therefore averaged to one minute. During all experiments with gaseous reagents, no evidence of catalyst performance degradation was observed."**

[Figure]

**"Figure 4. Allan-Werle deviation (3σ) in the HCl-TCl purged with zero-air (black line) shown with the ideal deviation (no drift, solid blue line) and associated error in the deviation (dashed blue line)."**

SI:

**Table S3.** Response time of the HCl-TCl tested using for three chlorinated compounds.

| Tested TCl$_g$ species | Mixing ratio (ppbv) | Residence time (s) | $1/e$ (s) | $t_{90}$ (s) |
|---|---|---|---|---|
| Dichloromethane (DCM) | 164 | 1.5 | 23 | 189 |
| 1-Chlorobutane (CB) | 14.5 | 1.5 | 14 | 162 |
| 1, 3-Dichloropropene (CP) | 950 | 1.5 | 22 | 42 |

The method for quantifying response time is by calculating the e-folding response time (1/e) a 37% signal loss and t90 a 90% signal loss with respect to time in seconds.

(9) More explanation as to role of propane and inlet chemistry is needed. Propane auto-ignites at a temperature 470 °C. Here, it is being heated to >800 °C. What does this mean for the proposed mechanism in which HCl is formed from reaction of Cl atoms with propane (R1)? It is certainly not as simple as suggested here. More likely, HCl is formed as a result of a series of combustion reactions involving propane oxidation products. There is a lot of literature on the combustion of propane over platinum catalysts that should be incorporated in the discussion (e.g., Titova, N.S., Kuleshov, P.S. & Starik, A.M. Kinetic mechanism of propane ignition and combustion in air. Combust Explos Shock Waves 47, 249–264 (2011). https://doi.org/10.1134/S0010508211030014) . In particular, section 3.2 should be rewritten.

While the Reviewer is correct that propane auto-ignites at 470 °C, that is only when concentrations in air are above ~10%. This is far above the concentrations of propane that are provided in this system. Our proposed mechanism is a hypothesis, since a full exploration of any Pt-catalyzed mechanisms and related series of reactions that take place is beyond the scope of this work.

We have modified the text in the Introduction as follows:

"These Cl atoms readily form HCl via hydrogen abstraction (R1), in this case from propane **(or its thermolysis products)** that is supplied in excess."

Specific comments

Title / line 2: "In ambient air". One of the applications described is indoor air measurement. Consider broadening the scope of the title - for example, replace "in ambient air" with "in air".

Title has been changed to:

**"Development and Validation of a New In-Situ Technique to Measure Total Gaseous Chlorine in Air"**

line 15. "combusting ambient air". Most people would define combustion as the rapid chemical combination of a substance with oxygen, which is not the case here (at least not for the monitored HCl). 'Combusting' is a poor choice of words in this context. Please rephase.

Combusting has been replaced by "**thermolyzing**".

Is there an inlet aerosol filter before air enters the furnace? If not, HCl could be driven off atmospheric particles.

Experiments were conducted to explicitly test the conversion of particulate chloride in the HCl-TCl and are explained above.

A filter was not used in the ambient outdoor air sampling that had been reported. Though we expect particulate chloride to be very low during the sampling times that had been included (November and August), we have collected new ambient data and changed the text.

Section 2.5:

"Outdoor air sampling was performed between **00:00 on July 7 to 20:00 on July 11, 2022** (Eastern daylight time, EDT)…The outdoor air was pulled through a 2.5 μm particulate matter cut-off URG Teflon Coated Aluminum Cyclone (URG Corporation, Chapel Hill, North Carolina, USA) **to remove larger particles and then passed through a PTFE filter (2 μm pore size, 47 mm diameter, TISCH scientific, North Bend, Ohio, USA).**"

Section 3.4:

"The maximum, minimum, and median of observed ambient $TCl_g$ were **536.3, 2.0, and 3.1 ppbv, respectively (Figure 5)**. Measurements of HCl alone were not made during these periods but reported ranges of HCl mixing ratios for this sampling location from Furlani et al. (2021) and Angelucci et al. (2021) were typically below 110 pptv, with intermittent events up to 600 pptv. **As expected, most ambient $TCl_g$ measurements were above the expected mixing ratio of $LLCl_g$. There is clear evidence of $TCl_g$ sources beyond $LLCl_g$ at the sampling site, with several plumes of elevated $TCl_g$ intercepted. For example, the maximum $TCl_g$ measurement was made in a plume just after noon on July 7. Another plume was detected on July 11, with a maximum $TCl_g$ of 42.1 ppbv. Though the purpose of this study was not to determine sources of $TCl_g$, we observed that plumes containing elevated $TCl_g$ arrived from the S-SW of the sampling site, where several facilities that had reported tens to thousands of kg of yearly emissions to air of Cl-containing species are located (Figure S5).**"

[Figure]

"**Figure 5.** Monitoring meteorological conditions and **one-minute averaged** TCl$_g$ in outdoor air through HCl-TCl from July 7 to 11, 2022. Grey dashed line represents the background mixing ratio for LLCl$_g$."

Supplement:

[Figure]

"**Figure S5. (A) Conditional Probability Function (CPF) analyses of measured TCl$_g$, presented in as a polar plot, which displays the number of events when the concentration was greater than the 95th percentile as a function of both wind speed and direction. As a result, CPF polar plots present the probability that high concentrations of a pollutant came from a particular wind direction and speed and can give information on the contributions of local and regional sources** (Uria-Tellaetxe and Carslaw, 2014). **Polar plots were plotted in R using the openair function** (Carslaw and Ropkins, 2012). **(B) Map of sampling site (red star) and three facilities (purple dots) within 20 km of the site in the S-SW direction that reported release of 75 - 2700 kg of Cl-containing species to air in 2021**

**to the Canadian National Pollutant Release Inventory** (National Pollutant Release Inventory, 2022)."

line 18. "HCl, for which detection by CRDS has been shown to be fast and reliable" This statement is not appropriate to appear in this section (which is the abstract to this paper) because reliable and fast detection of HCl by CRDS was not demonstrated in this manuscript (it was published in an earlier manuscript instead).

Changed for clarity to:

"HCl, for which detection by CRDS has **previously** been shown to be fast and reliable."

line 19. Consider stating the criteria guiding the selection of this compounds (high BDE).

We have clarified the abstract and incorporated the information from the additional experiments. It now reads as follows:

"Complete conversion **was observed for six organochlorine compounds, including alkyl, allyl, and aryl C-Cl bonds, which are amongst the strongest Cl-containing bonds**. The quantitative conversion of these strong C-Cl bonds suggests complete conversion of similar or weaker bonds that characterize all other $TCl_g$."

line 22-24. Please state what quantities are plotted against each other - what is the reference measurement or benchmark against which the new instrument was evaluated?

Changed to:

"Complete conversion **of gravimetrically validated PDs** was indicated by the near unity orthogonal distance regression analysis slope ($\pm\sigma$) **of measured $TCl_g$ plotted against expected $TCl_g$ and was** $0.996 \pm 0.012$, $1.048 \pm 0.060$, and $1.027 \pm 0.061$ for DCM, CB, and DCP, respectively."

What is the meaning of the errors stated? Clearly, it is precision (since values > 1 are physically not feasible) - but at what level of confidence?

Here $\pm\sigma$ refers to $\pm$ 1 standard deviation in the weighted orthogonal distance regression slope calculation.

line 24 "Breaking these strong C-Cl bonds represents a proof of concept for complete conversion" replace with "The quantitative conversion of these test compounds that contain relatively strong C-Cl bonds suggests complete conversion"

Changed.

line 26-27 "reasonable comparisons" - one can compare apples and oranges. Suggest rephrasing to "reasonable agreement"

Changed to:

"… found reasonable **agreements** in ambient background mixing ratios…"

line 27 "sum of expected HCl from known Cl species" - expected how? known how? Be explicit.

Since this is the abstract, we have tried to be brief. However, we agree with the Reviewer that it could be clearer. We have modified this to:

"the sum of expected HCl from known **long-lived Cl** species."

This sum is expected to be highly variable in reality - while it's good to be in the right "ballpark", this comparison is not exactly a rigorous one.

We agree with the Reviewer. This is simply an additional real-world constraint to complement laboratory analyses. However, our reasonable agreement with expectations suggests a good conversion efficiency for the suite of chlorinated species found in the atmosphere.

line 61 - "unreactive chloride" - consider mentioning sea salt aerosol in this context.
This has now been added:

"…atmospheric processes transform relatively unreactive chloride (Cl$^-$, **such as sea salt, NaCl**) into reactive gaseous chlorine (Cl$_y$)…"

line 89 "Understanding TClg source and sink chemistry". The sources and sinks will differ for each component of TClg and are known in many cases. How does the measurement TClg fit in and help elucidate sources and sinks?

While it is true that sources and sinks will be different for each compound, by looking at TCl$_g$ in combination with meteorological data we can, for example, help to identify sources of any chlorinated compound by identifying source regions. These can then be further elucidated using speciated measurements.

line 118 - consider adding a table summarizing the various elemental measurements. Are they all using Pt catalyst?

We believe this is beyond the scope of this work. We agree that it would be useful to summarize these measurements, perhaps in a future review paper.

line 137 "Clean" - replace with "Chlorine-free"

This has been changed:

"**Chlorine-free zero** air was generated by a custom-made zero-air generator."

line 139 The (typical) CRDS sample flow rate should be stated [it is stated on line 203 but should be stated earlier].

Text has been added earlier to clarify:

"**The furnace transfer line met an overflow tee when delivering flows greater than the CRDS flowrate of 2 L min$^{-1}$.**"

line 148 "Spectral interference" What does this mean? Please explain and show the interfering spectra.

Under instances of high concentrations (10s ppm) of low absorbing compounds (i.e., propane) a false data flag is observed by the CRDS. This is due to unknown shifts and distortions from, in this case high levels of propane, to the normal HCl absorption profile that is used in the fitting algorithm. As we are unfamiliar with the actual fitting parameters used in the proprietary software, we simply label instances of highly concentrated and very lowly absorbing compounds as interferences. A more detailed explanation can be found in Furlani et. al. (2021).

line 149 "[propane] should only be added when temperatures exceed 650 °C [Furlani et al., 2021]" Furlani et al., 2021 do not mention propane or inlet furnaces, so more explanation as to this constraint is needed. Could it be that at temperatures above 650 °C most of the interfering propane is oxidized?

The Reviewer is correct, due to no observed false data flags from the CRDS after 650 °C we surmise that most of the propane is also undergoing either oxidation or thermolysis.

line 153. "CRDS can capture transient fast HCl formation processes on the time scale of a few minutes" Please show sample data to support this statement. What is the rise time?

Information on time response data is now included.

The new Section 3.3:

**"The response time of the instrument was assessed during experiments with DCM, CB, and CP. The time for the signal to decay after removal of the PDs was determined to 37 % (1/e) and 90 % ($t_{90}$) of the maximum signal. The maximum time to achieve 1/e was 23 seconds, while the maximum time to achieve $t_{90}$ was 189 seconds (Table S3). Given the high mixing ratios used to test the response times, we argue that under most conditions relevant to indoor and outdoor atmospheric chemistry, a sample integration time of one minute will minimize any time response effects. Data for outdoor and indoor sampling described in Sections 3.4 and 3.5 were therefore averaged to one minute."**

SI:

**Table S3.** Response time of the HCl-TCl tested using for three chlorinated compounds.

| Tested TCl$_g$ species | Mixing ratio (ppbv) | Residence time (s) | $1/e$ (s) | $t_{90}$ (s) |
|---|---|---|---|---|
| Dichloromethane (DCM) | 164 | 1.5 | 23 | 189 |
| 1-Chlorobutane (CB) | 14.5 | 1.5 | 14 | 162 |
| 1, 3-Dichloropropene (CP) | 950 | 1.5 | 22 | 42 |

The method for quantifying response time is by calculating the e-folding response time (1/e) a 37% signal loss and t90 a 90% signal loss with respect to time in seconds.

line 154. "limited by inlet effects" Would inlet effects? Please explain or rephrase.

Changed to:

"The coupled CRDS can capture transient fast HCl formation processes on the timescale of a few minutes, limited by the **high adsorptive activity of HCl on inlet surfaces**."

line 158 How is LOD defined ($1\sigma$, $2\sigma$ or $3\sigma$)?

This information is already provided as: "Limits of detection (LODs) were calculated as three times the Allan–Werle deviation in raw signal intensity when overflowing the inlet with zero air directed into the CRDS for $\sim$ 10 h." We have now added LODs for the HCl-TCl method as well (new Section 3.3).

line 160 Figure 1. Is there an overflow or tee between the furnace and the CRDS, or are both operated at the same flow rate?

Text has been added to clarify:

"**The furnace transfer line met an overflow tee when delivering flows greater than the CRDS flowrate of 2 L min$^{-1}$.**"

Are there any pressure drops at higher flow rates?
Pressure is measured and regulated in the CRDS cell ($\pm 0.0002$ atm); any pressure drops are flagged by the instrument.

lines 179-181 - does the 4-week period encompass the time period over which the work in the remaining manuscript was conducted, or was there extrapolation?

This encompassed the portion of time all three PDs were evaluated.

Some of these standard deviation are quite large, whereas others are very small. Please explain.

The molecules each permeate at different rates dues to steric and volatility properties. The diffusion through different-sized pores in the materials leads to different uncertainties for each.

line 190-191 "The conversion temperature was determined when the measured HCl plateaued at 100% conversion." Please rephrase for clarity.

Changed for clarity:

The **operating** temperature was determined **when complete conversion of the measured TCl$_g$ for the tested compounds was sustained** at 100% conversion **based on PD emission rates.**

line 192 "flows of 0.6-5.5 L min-1" - how was the furnace coupled to the CRDS or could the CRDS be operated at any of these flow rates if the CRDS flow rate was 2 L min-1 (line 203 - see also line 160). If the CRDS flow rate was variable, is the line shape/pressure of HCl affected?

Text has been added to clarify:

"**The furnace transfer line met an overflow tee when delivering flows greater than the CRDS flowrate of 2 L min$^{-1}$.**"

line 208. It sounds as if ambient air was sampled without an aerosol filter. Please comment on the conversion of aerosol chloride to HCl (e.g., evaporation of ammonium chloride).

See above text and figure S3.

line 225-227 Strike "Instances ... (Furlani et al., 2021)" as this is standard practice and certainly isn't unique to this group.

Reference has been removed.

line 229-236 Has the H2O2 CRDS been validated to quantify HOCl? More detail is needed to describe the HOCl measurement here (rather than refer to an unwritten/unpublished manuscript draft).

This instrument has been validated. The manuscript describing the validation has been submitted and the reference updated to reflect the submission of this manuscript. Unfortunately, AMT does not provide a mechanism for us to directly share the manuscript under review with Reviewers for their consideration.

**"Stubbs, A., Lao, M., Wang, C., Abbatt, J., Hoffnagle, J., VandenBoer, T., and Kahan, T.: Near-source hypochlorous acid emissions from indoor bleach cleaning, Environ Sci Process Impacts, Submitted, October 2022."**

line 246 "Test all TClg species is not feasible" - probably not, but surely more than 3 (and especially HCl) should have.

We have now examined HCl and three additional chlorine-containing species. Effective conversion of six Cl-containing compounds with high bond dissociation energies, encompassing alkyl, allyl, and aryl C-Cl bonds have now been tested. This is consistent with the number of compounds tested for validation of other methods. For example, Yang and Fleming (2019) demonstrated the conversion of six compounds in their demonstration of a Pt-based total carbon measurement.

line 259 "six residence times" does the pressure inside the furnace change when the residence time (flow rate) are altered?

Pressure in the flow tube was not monitored but with changing flowrate through the tube it is very likely.

line 263 "Expected TClg" - Clearly state what you mean by "Expected" - Is it what is summarized Table S1 under expected or based on the leak rates in section 2.3? Note my major concern #4 about using gravimetrically calibrated diffusion sources.

This has been revised for clarity:

"The temperature breakthrough was observed when complete conversion of the **expected HCl for tested compounds based on PD emission rate was stable after reaching the optimal temperature**. It was found to be ~800 °C for the tested organochlorines (Figure S2)."

line 264 "optimal" - please state the criteria that were considered optimal (highest conversion efficiency at fastest flow rate?). Based on Fig 2., I would have not thought 1.5 s would be optimal as the conversion efficiency was greater at 4.5 s.

Text added for clarity:

> "An **optimal** residence time of 1.5 seconds was selected for all HCl-TCl experiments **for its good conversion efficiency and relatively better response time (see Table S3).**"

line 273 "slower response time due to increased surface effects of HCl" Please show the data and state how slow the response was.

Text added:

> "measured HCl suffered from longer equilibration times **(~30 minutes, more than double the 1.5 residence time)** and therefore a slower response time, **likely** due to increased surface effects of HCl after exiting the furnace."

Figure 2. Please change the axis caption to SI units (seconds should be s).

Updated

The figure caption should state relevant details such as oven temperature, flow rate, propane concentration, pressure, etc.

Text has been added to clarify:

> "Conversion efficiency of DCM plotted against residence time in the HCl-TCl **at 825 °C.**"

Is the scatter shown in Figure 2 what one would expect (i.e., are the dips at 0.9 s and 2.6 s reproducible or indicative of some sort of random error?). Could you draw a smooth line through all the data?

Rather than showing the run-time precision as error bars, consider repeating the experiment several times over several days and showing the day-to-day precision (which is likely larger).

While we appreciate the Reviewer's point that this experiment could be repeated many times to produce a more realistic assessment of the error, the purpose of this experiment was not to fully understand the relationship between residence time and response. Rather, our goal was to determine a residence time at which we could expect reasonable results. Thus, we chose not to spend additional time and resources to better constrain the error. Our additional conversion experiments serve to validate the choice of residence time. The working residence time may in fact be a range, but for the sake of brevity we went with the lowest working residence time (i.e., 1.5 seconds).

Text was added to Figure 2 caption:

> "**Note that the error bars are represented by the precision of the instrument, and we expect there would be greater experiment-to-experiment variability.**"

line 283-285 - "The mixing ratios tested for DCM were 41, 54, 80, 111, 284 and 165 ppbv. The mixing ratios tested for CB were 3.5, 4.6, 6.8, 9.5, and 14 ppbv. The mixing 285 ratios tested for DCP were 121, 259, 468, 651, and 967 ppbv." Please rectify the poor sentence flow and clearly

state how these mixing ratios were determined or calculated. If they are based on the mass emission rates from line 180-181, uncertainties/error should be added.

The text was changed and information was added to the SI:

**"See table S4 for summary of mixing ratios used, all lower mixing ratios were generated by diluting the highest mixing ratio of each compound by chlorine-free zero air."**

**Table S4**. Summary of mixing ratios used for HCl-TCl conversion efficient experiments.

| DCM mixing ratio (ppbv) | CB mixing ratio (ppbv) | DCP mixing ratio (ppbv) |
|---|---|---|
| 41± 0.6 | 3.5 ± 0.6 | 121 ± 2 |
| 54 ± 0.8 | 4.6 ± 0.8 | 259 ± 4 |
| 80 ± 1 | 6.8 ± 1 | 468 ± 8 |
| 111 ± 2 | 9.5 ± 2 | 651 ± 11 |
| 165 ± 3 | 14 ± 2 | 967 ± 16 |

Figure 3 - Axis title: "Expected normalized HCl from PD output": What does this mean?

Due to the differing emission rates of the PDs for each compound, we normalized the data for each compound to the highest added mixing ratio so they could be displayed together. For example, the highest mixing ratio for DCP was close to 1 ppmv, while the highest concentration for CB was ~25 ppbv.

Mixing ratio data for each compound is now shown in the SI:

[Figure]

**"Figure S4.** HCl-TCl measured HCl plotted against the expected HCl from converted DCM (A), 1,3-dichloropropene (B), and Chlorobutane (C) under three conditions. HCl-TCl conversion is shown for both Pt and propane added (black), with only Pt (dark grey), and only propane (light grey). Error bars on the y-axis represents 1σ in the HCl signal for 10 minutes. Error bars on the x-axis represent the error in the PD."

Please use different symbols (squares, circles and triangles) for DCM, CB and DCP.

Please do not dash the 1:1 line (it's a line, not a dash) and remove the straight fit lines for each of the standards as they obscure the data points.

These changes have been made to Figure 3:

[Figure]

"**Figure 3**. HCl measured by CRDS plotted against the expected HCl from HCl-TCl converted DCM (red **circle**), 1-chlorobutane (blue **diamond**), and 1,3-dichloropropene (green **square**) under condition (1). All values are normalized to the highest expected HCl concentration to better illustrate deviations from unity (black line). Error bars on the y-axis represent 1σ in the HCl signal over 10 minutes. Error bars on the x-axis represent the uncertainty in the PD used to generate DCM."

What is meant by "normalized" here? Why not plot concentration of HCl observed vs concentration of TClg emitted?

See above text for a description of the normalization. The data in terms of mixing ratio is now shown in the SI (see above).

Table 1 - Fix the formatting of the table (borders) to journal standards. Rather than defining conditions, simply state w/ Pt, w/o Pt, w/ C3H8, w/o C3H8 etc. in the table heading to improve manuscript clarity and readability.

The table has been edited as suggested by the Reviewer:

**Table 1.** Conversion efficiency for tested Cl-containing compounds under different conditions (both Pt and propane; Pt only; propane only). Note that chlorobenzenes were only tested under final Pt and propane conditions.

| Tested TCl$_g$ species | Cl bond dissociation energy (kJ mol$^{-1}$) | Conversion efficiency (%) | | |
|---|---|---|---|---|
| | | Pt and propane | Pt only | Propane only |
| Dichloromethane (DCM)[a] | 310 | $99.6 \pm 3.2$ | $80.7 \pm 2.4$ | $94.4 \pm 6.6$ |

| | | | | |
|---|---|---|---|---|
| 1-Chlorobutane (CB)[a] | 410 | 104.8 ± 5.6 | 54.1 ± 6.6 | 44.2 ± 5.9 |
| 1, 3-Dichloropropene (DCP)[a] | 350 | 102.7 ± 7.8 | 54.3 ± 5.2 | 41.7 ± 5.1 |
| Trichlorobenzene (TrCB)[b] | 400 | 97.0 ± 19.9 | | |
| Tetrachlorobenzene (TeCB)[b] | 400 | 90.6 ± 10.3 | | |
| Pentachlorobenzene (PeCB)[b] | 400 | 90.2 ± 14.8 | | |

[a]Conversion efficiency was determined from the orthogonal distance regression slope and ± σ and propagated error from individual permeation devices.

[b]Conversion efficiency was determined directly by the quantity (mol) of HCl measured from liquid injections of 1 mM standards. The error represents ±σ of measurements for n = 5 (TrCB) or n = 4 (TeCB, PeCB) injections.

line 328. "...expected to be between 3.3 and 19 ppbv". That's a huge range.

We agree with the Reviewer and believe this motivates the utility of a $TCl_g$ measurement. More information is certainly needed to better understand chlorinated chemicals in the atmosphere.

line 334 "chloride ... is assumed to not be converted". That's an easy experiment to verify. I wouldn't be surprised if ammonium chloride aerosol is quantitatively converted.

As described above, experiments were undertaken with particulate chloride in the form of NaCl. This confirmed the need for filters in-line prior to $TCl_g$ measurements, which is now emphasized in the text.

line 338 "Measurements of HCl alone were not made" - one could have alternated between furnace and ambient air using a simple switching valve.

While a valve is easy to include, the measurement of two very different concentrations of HCl is not so simple to switch. As a highly surface-active molecule, there is a lag time that would have greatly reduced the amount of quality data for $TCl_g$ that would have be obtained. Given that measurements have been made from the same location and shown to be a small fraction of the measured $TCl_g$, we did not believe this was a necessary experiment.

Figure 4. Interesting trend and variability in concentration. Can you prove that the signal does not change because of a variable conversion efficiency? I am suggesting a simple experiment adding a known amount (e.g., from one of your diffusion sources) hourly while sampling ambient air to verify that the conversion efficiency does not change.

Our replicate experiments showed no evidence of variable conversion efficiency, nor has this been observed for other total elemental measurements (e.g., Stockwell et al. (2018)).

Figure 5. SI units in axis caption, please. It is unclear what shade corresponds to which application.

The figure has been revised for clarity. The figure also now shows 1-minute averaged data, as a result of the response time analysis that is now included in the manuscript. This is reflected in the x-axis (now in min) and in the caption.

[Figure]

"**Figure 6. One-minute average** HCl (purple), HOCl (orange), and TCl$_g$ (black) observed during cleaning spray events. Mixing ratios were background corrected prior to each cleaning event. Each subsequent application of cleaner is illustrated by a lighter shade for HOCl and TCl$_g$."

Could you compare the response of the new instrument to HOCl with the CRDS (not sure if a source of pure HOCl(g) could be constructed)?

Unfortunately, a standard of pure HOCl cannot be generated. This is a well-established problem in determining atmospheric levels and fate of HOCl (e.g., Schwartz-Narbonne et al. (2018)), Wang et al. (2019)).

As described above, the paper describing the HOCl instrument is submitted.

line 410 "using DCM" Would you expect optimized conditions to be different for other molecules?

We chose DCM as it has a relatively strong C-Cl bond. We determined the most optimal conditions we could find, and if these conditions gave us poor results for the subsequent experiments with the other compounds, we would have revised the approach.

line 424 insert "and" between "Foundation" and "Natural" and remove comma

Changed.

line 433 - References.

The formatting for doi's should be consistent but varies between references.

Changed.

Please properly subscript CO2, SO2 etc. (line 458).

Changed.

The reference on line 476 has been published in AMT (remove "Discuss.").

Changed.

 Supporting information

Table S1 - the formatting of this table (borders, shading) is not to scientific standards.

The table has been modified.

 "observed" in the Table title is unnecessarily confusing (observed where? when? how?) and should be a range, not a single value. Suggest replacing with "Typical" or "Average" or "Expected".

Changed to simply "mixing ratio". The footnotes of the table indicate the source of the data for each chemical.

The table omits a number of chlorine containing compounds, for example, cyanogen and acid halides, and chlorinated aromatic compounds such as polychlorinated biphenyls (see major comments). For example, the CRC handbook lists the BDE of Cl-Na as 412 kJ/mol, Darwent, "Bond Dissociation Energies in Simple Molecules" NSRDS-NB Vol 31 (1970) gives a BDE for ClCN->Cl+CN of 439 kJ/mol, the CRC handbook list Cl-CN at 422.6 kJ/mol Cl-CF=CF2 at 434.7 kJ/mol, and Cl-C6H5 at 406.4 kJ/mol.

To our knowledge, these compounds have either never been reported in the atmosphere (e.g., cyanogen halides) or are present at ultra-trace levels (e.g., PCBs, present in the gas phase on the order of $10^{-3}$ pptv levels, (Mandalakis and Stephanou, 2002)). Therefore, there contributions to $TCl_g$ either cannot be assessed or are insignificant.

Figure S1 caption: "High spectral interference" The figure does not show a spectrum, let alone a spectral interference.

The caption has been edited to read:

> **"Figure S1. $TCl_g$ signal derived at different temperatures from thermolysis of DCM**. High **signal related to** spectral interference of added propane at low temperatures (<650 °C). $TCl_g$ (red) and temperature (black) during a typical ramping program. Propane disconnected immediately after interference observed to **preserve instrument integrity**."

Figure S1 axis - expand the y-axis range from to 0 to ~20 ppbv.

This has been changed, as has the label of the large peak resulting from a signal related to spectral interference.

[Figure]

"**Figure S1. TCl$_g$ signal derived at different temperatures from thermolysis of DCM. High signal related to** spectral interference of added propane at low temperatures (<650 °C). TCl$_g$ (red) and temperature (black) during a typical ramping program. Propane disconnected immediately after interference observed to **preserve instrument integrity**."

The signal appears to be increase / slope upwards between 11:45 am 12:50 pm. Explain how you concluded that thermolysis is complete.

We concluded that thermolysis was complete by comparing the averaged data over the last 30 minutes with the expected HCl emissions from the DCM PD within the uncertainty expected from instrument fluctuations. While the signal does increase during that time, it also decreases in the last minute and if given time would likely rise again within the known uncertainty. The efficacy of the temperature chosen was further validated through the remaining experiments described in the work.

Figure S2 caption: Please give relevant experimental details such as flow rate, DCM concentration, matrix (ambient air, filtered air, room air, etc.), was propane added and if so, how much, etc.

Text added:

"Figure S2. Monitoring DCM conversion from 300–800 °C. **Flow rate was ~2 L min-1, DCM mixing ratio was 165 ± 3 ppbv in chlorine free zero air. Propane and Pt catalyst were added as described in Section 2.2.**"

What happens when the temperature is increased above 800 °C?

We did not explore beyond 800C, since we observed full conversion at this temperature.

Figure S3. It's hard to see what's what here.

This figure has been revised to separate the three compounds as well as to show mixing ratios (rather than normalized data).

[Figure]

"**Figure S4.** HCl-TCl measured HCl plotted against the expected HCl from converted DCM (A), 1,3-dichloropropene (B), and Chlorobutane (C) under three conditions. HCl-TCl conversion is shown for both Pt and propane added (black), with only Pt (dark grey), and only propane (light grey). Error bars on the y-axis represents $1\sigma$ in the HCl signal for 10 minutes. Error bars on the x-axis represent the error in the PD."

Figure S4. Unclear what information this figure is adding to describe the new method. Consider saving this figure for another manuscript.

We disagree with the Reviewer. Inclusion of the decay of $TCl_g$ with respect to air exchange rate is an important aspect here. Comparison of decay of chlorinated species to air exchange rate is commonly included in studies of indoor bleach (e.g., Wong et al., (2017), Mattila et al., (2020)).

S1. References. Strike "Receive" from Crisp et al.

This has been done.

**References**

Carslaw, D. C. and Ropkins, K.: openair — An R package for air quality data analysis, Environmental Modelling & Software, 27–28, 52–61, https://doi.org/10.1016/j.envsoft.2011.09.008, 2012.

Fernando, S., Jobst, K. J., Taguchi, V. Y., Helm, P. A., Reiner, E. J., and McCarry, B. E.: Identification of the Halogenated Compounds Resulting from the 1997 Plastimet Inc. Fire in Hamilton, Ontario, using Comprehensive Two-Dimensional Gas Chromatography and (Ultra)High Resolution Mass Spectrometry, Environ Sci Technol, 48, 10656–10663, https://doi.org/10.1021/es503428j, 2014.

Furlani, T. C., Veres, P. R., Dawe, K. E., Neuman, J. A., Brown, S. S., VandenBoer, T. C., and Young, C. J.: Validation of a new cavity ring-down spectrometer for measuring tropospheric gaseous hydrogen chloride., Atmos Meas Tech, 14, 5859–5871, https://doi.org/https://doi.org/10.5194/amt-2021-105, 2021.

National Pollutant Release Inventory: https://www.canada.ca/en/services/environment/pollution-waste-management/national-pollutant-release-inventory.html, last access: 9 October 2022.

Mandalakis, M. and Stephanou, E. G.: Study of atmospheric PCB concentrations over the eastern Mediterranean Sea, Journal of Geophysical Research: Atmospheres, 107, ACH 18-1-ACH 18-14, https://doi.org/10.1029/2001JD001566, 2002.

Mattila, J. M., Lakey, P. S. J., Shiraiwa, M., Wang, C., Abbatt, J. P. D., Arata, C., Goldstein, A. H., Ampollini, L., Katz, E. F., Decarlo, P. F., Zhou, S., Kahan, T. F., Cardoso-saldan, F. J., Ruiz, L. H., Abeleira, A., Boedicker, E. K., Vance, M. E., and Farmer, D. K.: Multiphase chemistry controls inorganic chlorinated and nitrogenated compounds in indoor air during bleach cleaning, Environ Sci Technol, 54, 1730–1739, https://doi.org/10.1021/acs.est.9b05767, 2020.

Mitchell, G. D.: A REVIEW OF PERMEATION TUBES AND PERMEATORS, Separation and Purification Methods, 29, 119–128, https://doi.org/10.1081/SPM-100100005, 2000.

Scaringelli, F. P., O'Keeffe, A. E., Rosenberg, Ethan., and Bell, J. Paul.: Preparation of known concentrations of gases and vapors with permeation devices calibrated gravimetrically, Anal Chem, 42, 871–876, https://doi.org/10.1021/ac60290a012, 1970.

Schwartz-Narbonne, H., Wang, C., Zhou, S., Abbatt, J. P. D., and Faust, J.: Heterogeneous chlorination of squalene and oleic acid, Environ Sci Technol, 53, 1217–1224, https://doi.org/10.1021/acs.est.8b04248, 2018.

Seinfeld, J. N. and Pandis, S.: Atmospheric Chemistry and Physics: From Air Pollution to Climate Change, 2nd Ed., John Wiley & Sons, Inc., Hoboken, NJ, 2006.

Stockwell, C. E., Kupc, A., Witkowski, B., Talukdar, R. K., and Liu, Y.: Characterization of a catalyst-based conversion technique to measure total particulate nitrogen and organic carbon and comparison to a particle mass measurement instrument, 2749–2768, 2018.

Uria-Tellaetxe, I. and Carslaw, D. C.: Conditional bivariate probability function for source identification, Environmental Modelling & Software, 59, 1–9, https://doi.org/10.1016/j.envsoft.2014.05.002, 2014.

Wang, C., Collins, D. B., and Abbatt, J. P. D.: Indoor illumination of terpenes and bleach emissions leads to particle formation and growth, Environ Sci Technol, 53, 11792–11800, https://doi.org/10.1021/acs.est.9b04261, 2019.

WMO (World Meteorological Organization): Scientific Assessment of Ozone Depletion: 2018, Report No., Global Ozone Research and Monitoring Project, Geneva, Switzerland, 588 pp. pp., 2018.

Wong, J. P. S., Carslaw, N., Zhao, R., Zhou, S., and Abbatt, J. P. D.: Observations and impacts of bleach washing on indoor chlorine chemistry, Indoor Air, 27, 1082–1090, https://doi.org/10.1111/ina.12402, 2017.

Yang, M. and Fleming, Z. L.: Estimation of atmospheric total organic carbon (TOC) – paving the path towards carbon budget closure, Atmos Chem Phys, 19, 459–471, https://doi.org/10.5194/acp-19-459-2019, 2019.

---

## Author Comment (AC2)

We thank the Reviewer for their detailed and constructive comments. Please find our responses below in blue highlight, with changes to the manuscript indicated in **bold**.

General Comments:

Furlani et al. built a system to measure total gaseous chlorine (TClg) in ambient air. The system converts total chlorinated species to HCl using a heated platinum converter and measures the Cl content using an HCl analyzer. The conversion efficiency was validated using 3 organochlorine molecules. They examined the efficiency of the converter at different conditions, including conversion temperature and flow rates. They tested the system by applying it to measure both outdoor air and indoor air when cleaning with chlorine bleach. Overall, the paper is well written and presents a new method in measuring total chlorine in the atmosphere, which is valuable to the community. However, they should address the following major comments and a few specific comments.

Major comments:

The system was only evaluated for 3 organochlorine molecules, including dichloromethane, 1-chlorobutane, and 1,3-dichloropropene. These three molecules are relatively similar in structure, i.e., they are all chlorinated alkanes/alkenes. However, in the atmosphere, various chlorinated species (both organic and inorganic) are present, and they likely have different conversion efficiency to HCl in the system. They should conduct evaluation for more chlorinated species, e.g., chemicals with more diverse structures and properties. Furthermore, the authors should test the conversion efficiency for major inorganic chlorine species, such as Cl2, inorganic chloramines, HOCl, ClNO2, HCl, etc. These are major chlorinated species from indoor bleach cleaning (Mattila et al., 2020; Wong et al., 2017), and are important reactive chlorines in the ambient air.

We have added conversion efficiency tests to three aryl organochlorine compounds. Our conversion efficiency tests now encompass alkyl, allyl, and aryl organochlorines, with six separate compounds tested. The text and Table 1 have been updated to reflect these additional experiments. Unfortunately, calibration standards cannot be created for most inorganic Cl-containing compounds, which precludes our ability to test them in the HCl-TCl. This is an ongoing challenge in the community (e.g. Mattila et al., (2020)). However, we note that we did generate a mix of inorganic chlorinated species through the surface bleaching experiment. Thus, this set of measurements indicates that inorganic chlorinated species are being converted in the HCl-TCl.

Section 2.4:

**"We tested the HCl-TCl conversion efficiency for two different quantities of three chlorobenzenes (TrCB, TeCB, and PeCB). Due to their high boiling points, PDs of these compounds could not be prepared. Instead, small volumes of approximately 1 mM solutions of these compounds dissolved in toluene were directly introduced to the HCl-TCl while it was sampling room air. Room air measurements of HCl were consistently <100 pptv, therefore should not influence experimental measurements. With a short piece of tubing used as an inlet, 1 and 2 µL of each compound was injected onto the inner surface**

**of the tubing, which was heated to ~100 °C with a heat gun to facilitate volatilization. The resulting signals were integrated over a time period of 2.5 hours to obtain the total quantity of HCl detected by the CRDS, which was used to calculate conversion efficiency. To account for uncertainties in peak integration, a high and low peak area boundary was determined, with the average peak area taken for each injection. Duplicates of each injected quantity were performed, except for 1 µL TrCB, which was performed in triplicate."**

Section 3.2:

**"To further validate the HCl-TCl, the conversion efficiency of three aryl chlorine compounds were tested under the final operating conditions (i.e., Condition 1, in the presence of both Pt and added propane). The TCl$_g$ measured from the three aryl compounds was unity, within the uncertainty of the measurement (Table 1).**

The results for all **six** compounds show that the HCl-TCl is capable of complete conversion of mono and polychlorinated species on sp$^3$ and sp$^2$ carbons using the determined temperature and flow conditions. The complete thermolysis of the strong**est** C-Cl bond on the primary alkyl chloride (CB) demonstrates the efficacy of the HCl-TCl. Breaking these relatively strong C-Cl bonds, **with consistent conversion efficiency across alkyl, allyl, and aryl C-Cl bonds**, is a good proof of concept for complete conversion of all bonds of similar or weaker bond energies that characterize all other TCl$_g$."

**Table 1.** Conversion efficiency for tested Cl-containing compounds under different conditions (both Pt and propane; Pt only; propane only).

| Tested TCl$_g$ species | Cl bond dissociation energy (kJ mol$^{-1}$) | Conversion efficiency (%) | | |
| --- | --- | --- | --- | --- |
| | | **Pt and propane** | **Pt only** | **Propane only** |
| Dichloromethane (DCM)[a] | 310 | 99.6 ± 3.2 | 80.7 ± 2.4 | 94.4 ± 6.6 |
| 1-Chlorobutane (CB)[a] | 410 | 104.8 ± 5.6 | 54.1 ± 6.6 | 44.2 ± 5.9 |
| 1, 3-Dichloropropene (DCP)[a] | 350 | 102.7 ± 7.8 | 54.3 ± 5.2 | 41.7 ± 5.1 |
| **Trichlorobenzene (TrCB)[b]** | **400** | **97.0 ± 19.9** | | |
| **Tetrachlorobenzene (TeCB)[b]** | **400** | **90.6 ± 10.3** | | |
| **Pentachlorobenzene (PeCB)[b]** | **400** | **90.2 ± 14.8** | | |

[a]Conversion efficiency was determined from the orthogonal distance regression slope and ± σ and propagated error from individual permeation devices.
[b]Conversion efficiency was determined directly by the quantity (mol) of HCl measured from liquid injections of 1 mM standards. The error represents ±σ of measurements for n = 5 (TrCB) or n = 4 (TeCB, PeCB) injections.

Another related question: How did the authors evaluate potential loss of reactive chlorine species on the inlet and instrument surfaces?

Inlet losses for most TCl$_g$ compounds are relatively unimportant compared to potential inlet losses of the produced HCl. While inlet losses may reduce the slower response time, the heated surfaces minimise permanent losses to the inlet.

The introduction: In the current version, the authors focused on discussing the importance of chlorine in the atmosphere in the Introduction. They should focus more on the measurement

techniques of chlorine, especially if there are any total chlorine measurement techniques in the literature, rather than the discussion on the importance of chlorine in the atmosphere. This helps to put the study in the right context, i.e., "development of measurement techniques for chlorine in ambient air". Thus, I suggest the authors to rewrite the introduction of the paper.

We have provided a discussion of the existing techniques that have been used to understand total chlorine (see fourth paragraph of the introduction). These have primarily been focused on the condensed phase. We believe our discussion of chlorine chemistry and reactive chlorine in the atmosphere is necessary to motivate the development of our technique and, thus, should remain in the introduction.

Specific comments:

Can the instrument measure particle phase chlorine?

Yes, additional experiments were performed to access this and show that a filter before the HCl-TCl will reduce the affects. We have tested the impacts of particulate chloride and added relevant text to the manuscript, as well as a figure to the SI:

Text has been added to the methods (Section 2.4):

"To determine if there was any positive bias in the TClg, measurement from the conversion of particulate chloride (pCl$^-$), NaCl aerosols were generated by flowing 2 L min−1 of chlorine free zero air through a nebulizer containing a solution of 2% w/w NaCl in deionized water. The aerosol flow was then mixed with 1 L min−1 of chlorine free dry zero air to achieve a total flow of 3 L min−1, The HCl-TCl (2 L min-1) then sampled off this main mixing line. Chloride was added after monitoring background zero air levels. After ~3 hours of measuring the converted pCl$^-$, a PTFE filter (2 µm pore size, 47 mm diameter, TISCH scientific, North Bend, Ohio, USA) was added inline onto the inlet of the HCl-TCl."

Text has been added to the results and discussion (Section 3.1):

"Conversion of particulate chloride (pCl$^-$) was observed to take place in the HCl-TCl (Figure S3), but once a filter was introduced the signal returned to background levels. Thus, to capture only gaseous TCl$_g$ from samples that may contain particulate chloride, a particulate filter must be used."

[Figure]

"**Figure S3. Testing the impacts of added particulate chloride (pCl⁻) to the HCl-TCl. TClg was monitored while pCl⁻ is added (red vertical dashed line) and then a Teflon filter is added (blue vertical dashed line) to the HCl-TCl inlet. Added filter showed complete reduction of the pCl⁻ signal.**"

Line 143: what is the size of the platinum mesh? Would the amount of Pt catalyst and the size affect the conversion of Cl species? For example, does finer Pt provide more surface area for the conversion reaction?

We have provided the surface area within the text (134 cm$^2$). Had any conversion limitations be observed, one approach we would have taken would have been to increase the surface area.

Line 149: the authors mention that "all lines and fittings were made of perfluoroalkoxy (PFA)". Were there any issues to use the PFA fittings and lines at high temperatures (~ 650 C-800 C)?

The Reviewer raises a good point. The insulation kept most heat within the furnace; most heat transfer occurred through the sample gas flow and left the transfer tubing and fittings warm to the touch, but never above manufacturer's recommended working temperature.

Line 154: Please explain about "inlet effects".

This has been revised.

"The coupled CRDS can capture transient fast HCl formation processes on the timescale of a few minutes, **limited by the high adsorption activity of HCl on inlet surfaces (discussed further in Section 3.3).**"

Additional information about inlet response time is now included in Section 3.3. and the SI:

"**The response time of the instrument was assessed during experiments with DCM, CB, and CP. The time for the signal to decay after removal of the PDs was determined to 37**

% (1/e) and 90 % (t$_{90}$) of the maximum signal. The maximum time to achieve 1/e was 23 seconds, while the maximum time to achieve t$_{90}$ was 189 seconds (Table S3). Given the high mixing ratios used to test the response times, we argue that under most conditions relevant to indoor and outdoor atmospheric chemistry, a sample integration time of one minute will minimize any time response effects. Data for outdoor and indoor sampling described in Sections 3.4 and 3.5 were therefore averaged to one minute."

**Table S3.** Response time of the HCl-TCl tested using for three chlorinated compounds.

| Tested TCl$_g$ species | Mixing ratio (ppbv) | Residence time (s) | $^1/_e$ (s) | t$_{90}$ (s) |
|---|---|---|---|---|
| Dichloromethane (DCM) | 164 | 1.5 | 23 | 189 |
| 1-Chlorobutane (CB) | 14.5 | 1.5 | 14 | 162 |
| 1, 3-Dichloropropene (CP) | 950 | 1.5 | 22 | 42 |

The method for quantifying response time is by calculating the e-folding response time (1/e) a 37% signal loss and t90 a 90% signal loss with respect to time in seconds.

Figure 1: Add flow rates in the diagram. Where is the inlet position? Please add the sampling inlet location.

The figure has been updated.

[Figure]

**Figure 1.** Sampling schematic showing the key components of the HCl-TCl coupled to the CRDS analyzer. Dashed lines indicate parts of the apparatus used only during validation. Not to scale.

Line 177: "DryCal Definer" should be "DryCal Defender"

This is an older model to the Defender and is called Definer.

Session 2.4: Was the HCl-TCl optimized for "inorganic chlorine species"?

Providing calibrated sources of inorganic chlorine species is an ongoing challenge for the measurement of these species (e.g. Mattila et al., (2020)). Our confidence in the conversion efficiency is further boosted by the lower bond dissociation energies of inorganic compared to organic chlorinated species. While we did not generate calibrated standards of inorganic chlorine to test in our system, we generated a mix of inorganic chlorinated species through the surface bleaching experiment. Studies have shown that the chlorinated species emitted from these processes are primarily inorganic. Thus, this set of measurements indicates that inorganic chlorinated species are being converted in the HCl-TCl.

Line 203: CRDs flowrate of 2 L/min. Is this flow a subflow of the inlet flow? It would be helpful to specify the flows in the diagram in Figure 1.

Updated

Line 214-215: If the inlet lines and fittings were maintained at 20-25 C, which is lower than outdoor temperatures (25-28 C), was there water condensation when the humid air from outdoors (at higher T) come indoors (at lower T) into the instrument?

There was no condensation observed in the lines during ambient sampling. We have removed the references to temperature during the ambient sampling period.

Other than conversion temperature and flow rate, did the authors test the effect of water/humidity on conversion efficiency? And how does RH influence the ambient measurement? This is important for ambient air measurement when RH varies.

Relative humidity would likely not play a factor in the conversion efficiency due to the fact that even 100% humidity at 25 °C will amount to negligible humidity at 825 °C, given the saturation temperature of water at 825 °C. Humidity effects have been studied previously in the group for the HCl measurement using CRDS, see Furlani et. al., (2021). In addition, a previously-described Pt-based total carbon method (Yang and Fleming, 2019), did not see any variation with RH in their Pt-based total carbon method.

Line 218: a URG Teflon coated aluminum cyclone was used to remove particles?

Changed to:

> "The outdoor air was pulled through a **2.5 μm particulate matter cut-off** URG Teflon Coated Aluminum Cyclone (URG Corporation, Chapel Hill, North Carolina, USA) **to remove larger particles.**"

Line 247: please define what is "strong Cl-containing bonds". Is there a threshold for "strong" vs. "weak"?

The strong vs weak argument here is all just stronger or weaker relative to the HCl BDE the highest tested chlorine containing compound.

See changed text:

"… of **relatively** weaker Cl-containing bonds (Tables S1 and S2)."

Line 254: what is "breakthrough temperature"?

Text added to clarify.

"The temperature breakthrough was **observed when complete conversion of the expected HCl for tested compounds was stable after reaching the optimal temperature and was** found to be ~800 °C for the tested organochlorines (Figure S2)."

Line 256: how about the temperature for inorganic chlorine?

As described later on in the text, we did not test any inorganic chlorine directly in experiments. Unfortunately, generating pure standards for calibration of these compounds is difficult and/or impossible. From the relatively weaker Cl-containing bonds in inorganic species compared to the tested compounds, we infer these bonds will also break and yield good efficiency. We have indirect evidence that we had good conversion of inorganic Cl-containing chemicals through the measurement of the bleach application.

Line 283: these are very high levels. In the real ambient air, their mixing ratios are a lot lower.

Correct, but due to the difficulty of generating atmospherically relevant mixing ratios of these compounds using PDs we could not go any lower.

Line 289: why are some conversion efficiency >100%?

All measurements are within the uncertainty near 100%.

Line 290-292 are repeating the information on line 289

Text added for clarity:

"… both Pt and propane the HCl-TCl conversion was 99.6 ± 3.2, 104.8 ± 5.6, and 102.7 ± 7.8%   for DCM, CB, and DCP, respectively (Table 1**), as the average conversion efficiency ± relative standard deviation**. From Figure 3 the comparison between expected and measured TClg is illustrated by near unity in the orthogonal distance regression slope (±1σ**, the error in the regression analysis**), and was 0.996 ± 0.012, 1.048 ± 0.0060, and 1.027 ± 0.061 for DCM, CB, and DCP, respectively."

Line 330: why not test the effect of particle chloride on TCl measurement? The authors could test with chloride containing salt particles.

See above explanation.

Line 331-332: "the conditions required to convert chloride to chlorine atoms ..." Do the authors mean organic or inorganic chloride?

This was deleted and replaced with the results to our particulate chloride measurements (see above).

Line 343: "in the during the"- delete "in the" or "during the".

Changed.

Line 359: "productfour" should be "product four"

Changed.

Line 361 and 362: is it "pptv" or "ppbv"? Mattila and Wong et al. observed 100s ppb level, not ppt.

This was in pptv and was only in reference to work done by Dawe et. al. The following sentence that refers to Mattila and Wong is related to the fast increase and short-lived nature of the emitted chlorinated compounds.

Line 379: It is unclear what the authors meant - "there was on average 82% of integrated TCl for which we cannot account."

This has been rephrased:

"In our experiments, there was on average $82 \pm 4$ % of integrated TClg **that could not be accounted for by the HOCl measurement**."

**References**

Furlani, T. C., Veres, P. R., Dawe, K. E., Neuman, J. A., Brown, S. S., VandenBoer, T. C., and Young, C. J.: Validation of a new cavity ring-down spectrometer for measuring tropospheric gaseous hydrogen chloride. Accepted, Atmos Meas Tech, 14, 5859–5871, https://doi.org/https://doi.org/10.5194/amt-2021-105, 2021.

Mattila, J. M., Lakey, P. S. J., Shiraiwa, M., Wang, C., Abbatt, J. P. D., Arata, C., Goldstein, A. H., Ampollini, L., Katz, E. F., Decarlo, P. F., Zhou, S., Kahan, T. F., Cardoso-saldan, F. J., Ruiz, L. H., Abeleira, A., Boedicker, E. K., Vance, M. E., and Farmer, D. K.: Multiphase chemistry controls inorganic chlorinated and nitrogenated compounds in indoor air during bleach cleaning, Environ Sci Technol, 54, 1730–1739, https://doi.org/10.1021/acs.est.9b05767, 2020.

Yang, M. and Fleming, Z. L.: Estimation of atmospheric total organic carbon (TOC) – paving the path towards carbon budget closure, Atmos Chem Phys, 19, 459–471, https://doi.org/10.5194/acp-19-459-2019, 2019.

---

## Author Response (AR2)

We thank the Reviewer for their time in providing feedback on our revised manuscript. We have addressed their remaining comment below, where our responses are indicated in blue and changes to the manuscript in **bold**.

*Reviewer 2*

The authors have carefully addressed the previous comments. The paper is much improved and can be accepted after addressing the following comments:

They authors added test with filters to remove chloride containing particles, which can lead to increasing signals of the HCl instrument. However, many Cl-containing gaseous species are semi-volatile, and HCl itself is also semi-volatile. How to rule out the effect of artifacts lead by the filter?

> We agree with the Reviewer that use of a filter could lead to bias in the measurement. We have added text to the manuscript in two sections to address this.

> Section 3.2

> "Thus, to capture only gaseous $TCl_g$ from samples that may contain particulate chloride, a particulate filter should be used. **Use of a filter could introduce blow on (i.e., partitioning of semi-volatile species) and/or blow off (i.e., processing of particulate chloride) artifacts. We have previously shown that HCl—likely to be the most surface-active component of $TCl_g$—is not greatly impacted by the presence of filters (Furlani et al., 2021), indicating blow on effects are likely minimal. However, the extent to which blow on effects should be considered will depend on the composition of the $TCl_g$ mixture and the temperature. Blow off effects will depend on ambient particulate chloride levels and can be mitigated by regularly changing the filter to prevent significant particulate chloride accumulation.**"

> Section 3.4

> "**The filter present in the inlet was unlikely to have led to artifacts in this measurement. Particulate chloride is negligible in continental summertime environments (Kolesar et al., 2018), indicating blow off artifacts would be minimal. Most** ambient $TCl_g$ measurements were above the expected mixing ratio of $LLCl_g$. **It is possible that semi-volatile chlorinated species could have partitioned to the filter, acting as a blow on effect, and leading to an underestimate of $TCl_g$. However, the warm temperatures during sampling (13 to 31 °C) and high observed $TCl_g$ levels suggest this was not a large effect.**"

*Other changes*

> We have added reference to the datasets that can now be found in the Federated Research Data Repository (Furlani et al., 2022). This has been referenced in the main text and in the Data Availability section.

References

Furlani, T., Ye, R., Stewart, J., Crilley, L., Edwards, P., Kahan, T., and Young, C.: Outdoor and indoor gaseous total chlorine measurement in Toronto Canada [data set], Federated Research Data Repository, https://doi.org/10.20383/103.0649, 2022.

Kolesar, K. R., Mattson, C. N., Peterson, P. K., May, N. W., Prendergast, R. K., and Pratt, K. A.: Increases in wintertime PM2.5 sodium and chloride linked to snowfall and road salt application, Atmos Environ, 177, 195–202, https://doi.org/10.1016/j.atmosenv.2018.01.008, 2018.